# Local activity alterations in individuals with autism correlate with neurotransmitter properties and ketamine-induced brain changes

Pascal Grumbach [1,2], Jan Kasper[1,3], Joerg F. Hipp [4], Anna Forsyth[5], Sofie L. Valk [1,3,6,7], Suresh Muthukumaraswamy [5], Simon B. Eickhoff[1,3], Leonhard Schilbach [8,9] & Juergen Dukart [1,3] ✉

Autism is a neurodevelopmental condition associated with altered resting-state brain function. An increased excitation-inhibition ratio is discussed as a pathomechanism but in-vivo evidence of disturbed neurotransmission underlying functional alterations remains scarce. We compare local resting-state brain activity and neurotransmitter co-localizations between autism (N = 405, N = 395) and neurotypical controls (N = 473, N = 474) in two independent cohorts and correlate them with excitation-inhibition changes induced by glutamatergic (ketamine) and GABAergic (midazolam) medication. Autistic individuals exhibit consistent reductions in local activity, particularly in default mode network regions. The whole-brain differences spatially overlap with glutamatergic and GABAergic, as well as dopaminergic and cholinergic neurotransmission. Functional changes induced by NMDA-antagonist ketamine resemble the spatial pattern observed in autism. Our findings suggest that consistent local activity alterations in autism reflect widespread disruptions in neurotransmission and may be resembled by pharmacological modulation of the excitation-inhibition balance. These findings advance understanding of the neurophysiological basis of autism. Trial registration number: ACTRN12616000281493

Autism is a neurodevelopmental condition primarily characterized by difficulties in communication, differed social interaction and presence of stereotyped and repetitive behavior. Diagnosis is based on behavior and frequently complicated by the heterogeneity of symptoms as well as the overlap with other psychiatric conditions[1,2]. The underlying neurobiological mechanisms of autism remain poorly understood, reliable biomarkers are lacking, and as a result, an effective pharmacological treatment model has yet to be established (for those who experience it as a source of suffering)[3].

Resting-state functional magnetic resonance imaging (rs-fMRI) has the potential to offer objective insights into the neurophysiology of autism. Therefore, numerous neuroimaging studies have investigated resting-state functional alterations in individuals with autism compared to typically developed controls (TD). Despite variability in these findings, potentially due to the use of different measures, investigations of local activity measures (e.g., ReHo) and meta-analyses have identified consistent patterns of both functional hyperactivity and -connectivity in the cerebellum, temporal and sensorimotor areas,

alongside hypoactivity and -connectivity within default mode network (DMN) nodes - a network important for self-referential processing and social cognition[4–12]. To date, little is known about specific neurotransmitter properties underlying this functional reorganization in autism and how these alterations relate to specific symptom domains[13–15]. Emerging evidence has implicated the dopaminergic modulation of motor pathways in manifestation of stereotyped behavior[16,17]. Furthermore, increased serotonin reuptake has been linked to atypical social behavior and sensory development in a subset of children with autism[18].

The imbalance of excitation and inhibition (E/I) is a common mechanism discussed in relation to autism. In line with that, alterations in glutamatergic neurotransmission have been proposed as pivotal in the neurophysiology of autism[19,20]. For example, Galineau et al. (2022) reported an overall increased density of the metabotropic glutamate receptor mGluR5 in male adults with autism[21]. Several studies support the hypothesis that N-Methyl-D-Aspartate (NMDA) receptor dysfunction contributes to autistic symptoms indicating that these symptoms can be improved by the NMDA receptor agonist D-Cycloserine[19,22]. Moreover, Siegel-Ramsey et al. (2021) found a negative association between increased glutamate concentrations in the dorsal anterior cingulate cortex (ACC) and reduced functional connectivity between the dorsal ACC and insular, limbic and parietal regions in male participants with autism[23]. Findings of altered glutamate neurotransmission, coupled with evidence of GABAergic (γ-aminobutyric acid) synaptic dysregulation in autism[24,25], have converged to support the hypothesis of an elevated E/I ratio as a potential molecular model for autism[26,27]. Despite an increasing body of evidence identifying autism-related molecular and functional alterations, the relationship between both types of changes remains poorly understood. A viable way of addressing this question may be by modifying the E/I balance using glutamatergic or GABAergic drugs to modulate either excitation or inhibition respectively. One could then directly test whether such modulation induces functional changes that are similar to alterations observed in autism providing in-vivo evidence on the similarity of both effects[28–30].

The aim of the present study was to better understand the neurochemical basis of functional alterations in autism by probing for their co-localization with in-vivo derived neurotransmitter information. We further test for similarity of autism-related alterations with ketamine and midazolam induced functional brain changes. We hypothesize that (1) autistic individuals show consistent local hyperactivity in sensorimotor and hypoactivity in DMN areas that co-localize with the spatial distribution of specific neurotransmitters and that (2) ketamine and midazolam induced neurochemical patterns are similar to these autism-related alterations. More specifically, based on evidence from previous studies, we anticipate the strongest co-localizations with glutamatergic, GABAergic, serotonergic and dopaminergic neurotransmission[14,31]. To improve the robustness of our findings, all hypotheses were initially tested using the ABIDE1 dataset and subsequently validated in the independent ABIDE2 dataset[32].

## Results

### Autistic individuals across datasets differ regarding age and symptom severity

In both datasets, individuals with autism and TD controls did not differ in terms of age (Table 1). However, TD controls included a significantly higher proportion of female subjects. In ABIDE2, TD controls showed higher FIQ values as compared to autistic individuals, whereas there was no FIQ difference in ABIDE1. Moreover, individuals with autism showed more head motion during image acquisition than TD controls. Autistic individuals in ABIDE1 were older and exhibited a higher severity of autistic symptoms as compared to those in ABIDE2.

### Consistent local activity alterations in autism

First, we tested for autism-related alterations in local synchronization (LCOR) compared to TD controls. In whole-brain voxel-wise analyses in the ABIDE1 dataset, we found a pattern of LCOR reductions in autism in brain regions comprising DMN hubs (bilateral posterior cingulate cortex (PCC), precuneus, frontal pole, frontal medial cortex), ACC, paracingulate gyrus and precentral gyrus as well as right hemispheric temporal pole, insular and opercular cortex (Fig. 1A, and Supplementary Table S1, S2). Increases in LCOR were observed in autism in bilateral temporal regions, the cerebellum and right lateral occipital cortex and angular gyrus. The decrease but not the increase findings were largely replicated in ABIDE2 (Fig. 1A, and Supplementary Table S3, S4). The voxel-wise findings were robust but spatially less extended following changes in preprocessing pipeline including removing the first four initial scans, smoothing and gray matter signal regression (Supplementary Figs. S1–2). The unthresholded ABIDE1 and ABIDE2 LCOR $t$-contrast maps displayed a strong positive spatial association $(r(117) = 0.597, p < 0.001, 95\% \ CI = [0.467, 0.701])$

## Table 1 | Demographic and clinical characteristics of the ABIDE1 and ABIDE2 cohorts

| | ABIDE1[1] | | | ABIDE2[1] | | | |
|---|---|---|---|---|---|---|---|
| | Autism | TD | p (Autism vs. TD) | Autism | TD | p (Autism vs. TD) | p (Autism vs. Autism from ABIDE 1 and 2) |
| n | 405 (46.1) | 473 (53.9) | | 395 (45.5) | 474 (54.5) | | |
| age, years | 17.3 (±8.6) | 17.3 (±7.9) | 0.965[2] | 15.6 (±9.8) | 15.1 (±9.3) | 0.457[2] | 0.008[2] |
| age range, years | 7.0-64.0 | 6.5-56.2 | | 5.1-62.0 | 5.9-64.0 | | |
| sex, female | 50 (12.3) | 84 (17.8) | 0.026[3] | 53 (13.4) | 140 (29.5) | <0.001[3] | 0.651[3] |
| FIQ | 108.5 (±15.2) | 108.7 (±14.0) | 0.873[2] | 107.1 (±15.7) | 115.1 (±12.6) | <0.001[2] | 0.184[2] |
| maximal motion | | | | | | | |
| - translation, mm | 1.0 (±0.6) | 0.9 (±0.6) | 0.041[2] | 0.9 (±0.6) | 0.9 (±0.6) | 0.364[2] | 0.083[2] |
| - rotation, degree | 0.9 (±0.6) | 0.9 (±0.6) | 0.193[2] | 1.0 (±0.7) | 0.9 (±0.6) | 0.41[2] | 0.314[2] |
| ADOS | 12.1 (±4.1) | 1.3 (±1.4) | <0.001[2] | 10.4 (±3.6) | 1.8 (±1.7) | <0.001[2] | <0.001[2] |
| - communication | 4.1 (±2.5) | 0.5 (±0.7) | <0.001[2] | 3.1 (±1.4) | 1.2 (±1.1) | <0.001[2] | <0.001[2] |
| - RSI | 8.2 (±3.1) | 0.8 (±1.0) | <0.001[2] | 7.1 (±2.5) | 0.6 (±1.0) | <0.001[2] | <0.001[2] |
| - SBRI | 2.3 (±2.3) | 0.1 (±0.3) | <0.001[2] | 1.6 (±1.4) | 0.1 (±0.3) | <0.001[2] | <0.001[2] |

[1]Numbers present either absolute numbers plus percent or mean plus standard deviation, [2]$t$-test (two-tailed), [3]$\chi^2$-test (two-tailed), *TD* typically developed controls, *ABIDE* Autism Brain Imaging Data Exchange, *FIQ* full scale IQ, *ADOS* Autism Diagnostic Observation Schedule, *RSI* reciprocal social interaction, *SBRI* stereotyped behaviors and restricted interests. maximal available clinical data for ABIDE-I ADOS ($n = 300$ with Autism), for ABIDE-II ADOS ($n = 238$ with Autism).

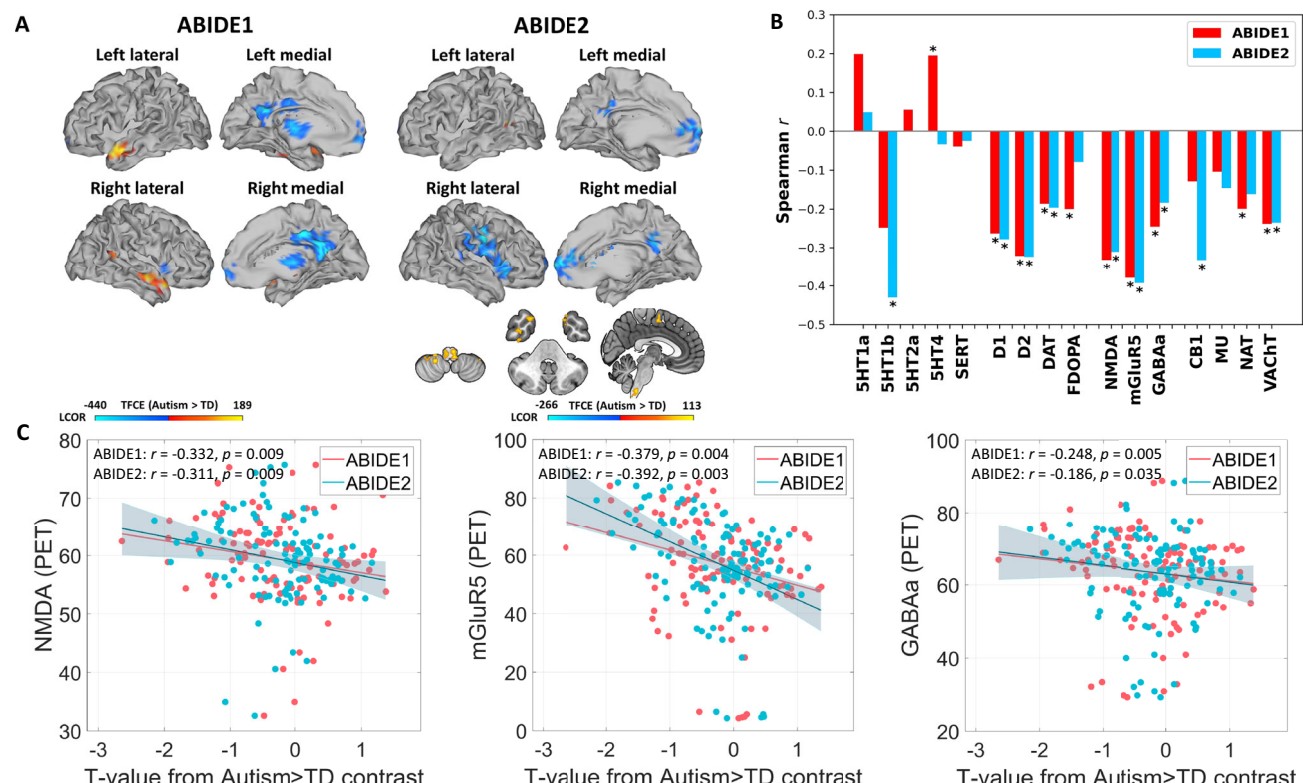

**Fig. 1 | Local functional activity alterations in subjects with autism and its neurotransmitter co-localizations. A** Results from voxel-wise comparisons between subjects with autism ($n = 405$, $n = 395$) and typically developed controls (TD; $n = 473$, $n = 474$) regarding local synchronization (LCOR). Positive T-values (red-yellow) indicate increased LCOR in subjects with autism compared to TD, negative T-values (blue) represent decreased LCOR. Left: Results from voxel-wise comparisons within the Autism Brain Imaging Data Exchange (ABIDE) 1 (sagittal view). Right: Results from voxel-wise comparisons within the replication dataset ABIDE2 (sagittal and axial view). Voxel-wise NIfTI files are available from the public repository (reference). **B** Grouped bar plot depicting the two-sided spatial co-localizations (Spearman correlations) of whole-brain LCOR alterations in subjects with autism compared to TD with 16 different receptor and transporter distributions. The asterisk (*) represents significant co-localizations ($p < 0.05$). Degrees of freedom (df) = 117. **C** Scatterplots showing the negative relationship between LCOR alterations in autism (T-values) and the nuclear imaging derived spatial NMDA, mGluR5 and GABAa receptor distributions. Trend lines represent the least-squares linear fit; shaded area denotes the 95 % confidence interval. Statistical analysis was performed using two-sided partial Spearman correlations (before multiple comparison correction). Df = 117. Abbreviations: 5-HT1a, 5-HT1b, 5-HT2a, 5-HT4 = serotonin receptor subtypes; D1, D2 = dopaminergic receptors; DAT dopamine transporter, SERT serotonin transporter, FDOPA dopamine synthesis capacity (fluorodopa PET), GABAa gamma-aminobutyric acid type A receptor, NMDA ionotropic N-methyl-D-aspartate receptor, mGluR5 metabotropic glutamate receptor subtype 5, MU μ-opioid receptor, CB1 cannabinoid receptor type 1, NAT noradrenaline transporter, VAChT vesicular acetylcholine transporter. Source data including exact $p$ values are provided as a Source Data file.

indicating a similar topology of autism-related alterations across both cohorts.

## LCOR alterations in autism co-localize with specific neurotransmitter systems

Next, we evaluated whether LCOR alterations observed in autism were spatially correlated to the distribution of specific neurotransmitter systems derived from nuclear imaging in healthy volunteers. Whole-brain LCOR alterations in subjects with autism in ABIDE1 were found to significantly co-localize with the spatial distribution of dopaminergic D1 ($r(117) = −0.263$, $p = 0.005$, 95% CI [−0.422, −0.087]) and D2 receptors ($r(117) = −0.322$, $p < 0.001$, 95% CI [−0.475, −0.151]) as well as DAT ($r(117) = −0.187$, $p = 0.033$, 95% CI [−0.356, −0.008]) and FDOPA ($r(117) = −0.200$, $p = 0.034$, 95% CI [−0.367, −0.021]). Further significant associations were observed with serotonergic 5HT4 receptors ($r(117) = 0.195$, $p = 0.024$, 95% CI [0.015, 0.362]), the ionotropic NMDA receptor ($r(117) = −0.332$, $p = 0.009$, 95% CI [−0.482, −0.161]), the metabotropic mGluR5 receptors ($r(117) = −0.379$, $p = 0.004$, 95% CI [−0.523, −0.214]), GABAa receptors ($r(117) = −0.248$, $p = 0.005$, 95% CI [−0.410, −0.072]), NAT ($r(117) = −0.202$, $p = 0.024$, 95% CI [−0.368, −0.023]) and the cholinergic transporter VAChT ($r(117) = −0.241$, $p = 0.003$, 95% CI [−0.404, −0.064]). The significant co-localizations

with D1, D2, DAT, NMDA, mGluR5, GABAa and VAChT were replicated in the ABIDE2 dataset (Fig. 1B, C, and Supplementary Table S5). The co-localization profiles were robust to changes in pre-processing pipeline including removing the first four initial scans and omitting smoothing and gray matter signal regression (see Supplementary Tables S6-7). The results were largely replicated when using the functional Schaefer + Melbourne/Tian parcellation (see Supplementary Table S8). The meta-analytically combined $p$ values from both cohorts survived the correction for multiple comparisons for all these replicated co-localizations (see Supplementary Table S9). In addition, the combined $p$ values for the associations with 5-HT1b, CB1 and NAT were also significant after correction for multiple comparisons.

## Ketamine- and midazolam-induced LCOR changes co-localize with neurotransmitter systems

Similarly to the effects of autism, we tested for co-localization of ketamine- and midazolam-induced changes (compared to placebo) with specific neurotransmitter systems. Ketamine administration induced LCOR changes significantly co-localized with D1 ($r(117) = −0.282$, $p\text{-FDR} = 0.032$, 95% CI [−0.434, −0.115]), NMDA ($r(117) = −0.338$, $p\text{-FDR} = 0.032$, 95% CI [−0.488, −0.173]) and GABAa receptors ($r(117) = −0.251$, $p\text{-FDR} = 0.037$, 95% CI [−0.402, −0.080])

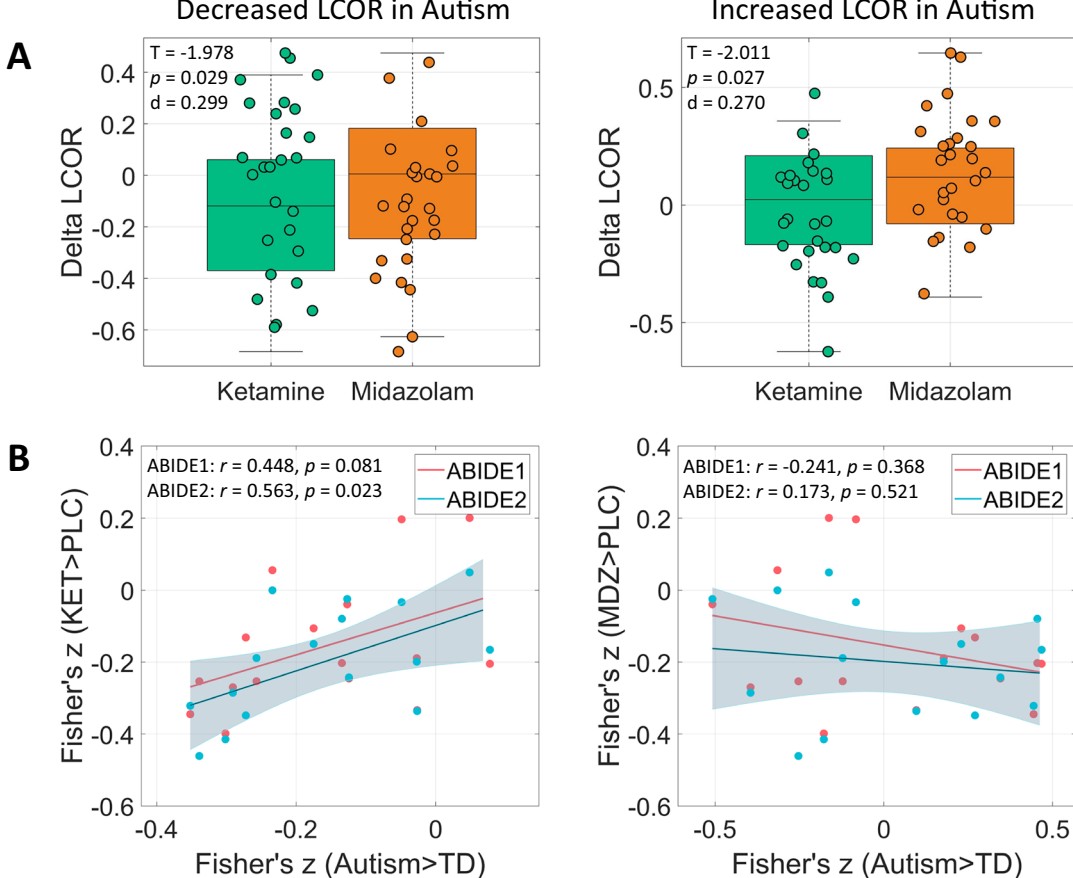

**Fig. 2 | Mean local activity (LCOR) after ketamine (KET) and midazolam (MDZ) administration in regions significantly altered in autism and correlation of the neurochemical co-localization profiles between autism, KET and MDZ condition. A** Boxplots comparing LCOR following KET and MDZ administration relative to placebo (PLC) condition. The data are derived from N = 27 independent individuals scanned in a three-way cross-over design, with each participant completing all three drug conditions (KET, MDZ, PLC). Delta LCOR is defined as the mean LCOR in the KET or MDZ condition minus the mean LCOR in the PLC condition. The plots display Delta LCOR values within all voxels that showed significant decreases (left) or increases (right) in individuals with autism compared to typically developing controls (TD) in the Autism Brain Imaging Data Exchange 1 (ABIDE1) dataset. Center line shows the median; box limits indicate the interquartile range (IQR; 25th–75th percentile); whiskers extend to the most extreme data points within 1.5 × IQR from the box. Statistical analysis was performed using one-sample t-tests (one-sided, $p < 0.05$). Degrees of freedom (df) = 26. **B** Scatterplots illustrating the relationship between the 16 Fisher's z-values for KET (left) and MDZ (right) versus PLC, and autism versus TD in the ABIDE1 and ABIDE2 datasets. The 16 Fisher's z-values represent the LCOR-neurotransmitter co-localization profiles for the respective contrast. Trend line represents the least-squares linear fit; shaded area denotes the 95 % confidence interval. Statistical analysis was performed using two-sided Pearson correlations. Df = 14. Source data including exact p values are provided as a Source Data file.

(Supplementary Table S10 and Supplementary Fig. S3). The effect of midazolam on LCOR co-localized with the distribution of 5HT2a ($r(117) = -0.305$, $p$-FDR $= 0.047$, 95% CI $[-0.452, -0.133]$), SERT ($r(117) = 0.436$, $p$-FDR $< 0.001$, 95% CI $[0.295, 0.557]$), DAT ($r(117) = 0.426$, $p$-FDR $= 0.002$, 95% CI $[0.283, 0.545]$), FDOPA ($r(117) = 0.417$, $p$-FDR $< 0.001$, 95% CI $[0.273, 0.537]$), CB1 ($r(117) = -0.468$, $p$-FDR $= 0.002$, 95% CI $[-0.588, -0.330]$), NAT ($r(117) = 0.264$, $p$-FDR $= 0.019$, 95% CI $[0.097, 0.423]$) and VAChT ($r(117) = 0.332$, $p$-FDR $< 0.001$, 95% CI $[0.178, 0.469]$).

### Autism-related alterations co-localize with ketamine effects
We tested for similarity of autism-related LCOR brain patterns with alterations induced by NMDA-receptor-antagonist ketamine and GABAa-potentiator midazolam (s. Supplementary Tables S11–14 and Supplementary Figs. S4–5 for detailed description and visualization of LCOR changes induced by ketamine and midazolam compared to placebo). Whole-brain LCOR alterations observed across both ABIDE datasets showed a significant positive correlation with ketamine-induced changes (ABIDE1: $r(117) = 0.381$, $p = 0.003$, 95% CI $[0.209, 0.530]$; ABIDE2: $r(117) = 0.292$, $p = 0.036$, 95% CI $[0.068, 0.475]$). The

effects of midazolam displayed no significant association with autism-related LCOR alterations (ABIDE1: $r(117) = 0.018$, $p = 0.903$, 95% CI $[-0.163, 0.198]$; ABIDE2: $r(117) = 0.205$, $p = 0.155$, 95% CI $[-0.038, 0.419]$). The ketamine and midazolam whole-brain t-contrast maps did not show a significant association (Supplementary Table S15).

We then tested for ketamine- and midazolam-induced LCOR changes in regions significantly altered in autism by computing t-tests for ΔKET and ΔMDZ changes (vs. placebo). Ketamine significantly reduced LCOR in regions showing a reduction in autism (Fig. 2A) ($T$ (26) $= -1.978$, $p = 0.029$, Cohen's $d = 0.299$, one-sided 90% CI $[-\infty, -0.039]$). Midazolam administration increased LCOR in regions displaying an increase in autism ($T(26) = 2.011$, $p = .027$, Cohen's $d = .270$, one-sided 90% CI $[0.036, \infty]$).

### Neurochemical co-localization profiles are similar between autism and ketamine conditions
Next, we tested for the overall similarity of the pharmacological neurotransmitter co-localization profiles with those observed in individuals with autism across all evaluated neurotransmitter systems. For this, we extracted mean Fisher's z values for the 16 tested receptors

and transporters for autism, ketamine and midazolam-related changes and computed Pearson correlations between these profiles. We found a significant association between the neurochemical profiles observed for ketamine and autism in ABIDE2 ($r(14) = 0.563$, $p = 0.023$, 95% CI [0.094, 0.828]) and a trend towards significance for ABIDE1 ($r(14) = 0.448$, $p = 0.081$, 95% CI [−0.061, 0.772]) (Fig. 2B). The neurochemical profile induced by midazolam did not show a significant association with the autism profile (ABIDE1: $r(14) = −0.241$, $p = 0.368$, 95% CI [−0.658, 0.288]; ABIDE2: $r(14) = 0.173$, $p = 0.521$, 95% CI [−0.353, 0.615]). Notably, the neurochemical autism profiles across ABIDE1 and ABIDE2 datasets were highly correlated ($r(14) = 0.814$, $p < 0.001$, 95% CI [0.534, 0.933]). The neurochemical profiles for ketamine and midazolam were not significantly correlated ($r(14) = 0.407$, $p = 0.118$, 95% CI [−0.110, 0.751]) (Supplementary Table S10).

### No consistent association of the LCOR-neurotransmitter co-localizations in autism with clinical symptom domains

Lastly, we tested for associations between individual z-scores of subjects with autism for the replicated whole-brain LCOR-neurotransmitter co-localization findings and different autism symptom domains as measured with the Autism Diagnostic Observation Schedule (ADOS)[32] (Supplementary Table S16). In ABIDE1, no significant associations were found with the ADOS total score nor the three subscores (communication, social interaction, stereotyped behavior and restricted interests [SBRI]). A significant but weak negative association was observed in ABIDE2 between SBRI and the strength of LCOR-GABAa co-localizations ($r(236) = −0.133$, $p = 0.040$, 95% CI [−0.255, −0.006]) and a positive with the strength of LCOR-VAChT co-localizations ($r(236) = 0.141$, $p = 0.030$, 95% CI [0.014, 0.264]) (Supplementary Fig. S6). Both findings did not survive a correction for multiple comparisons.

## Discussion

Here, we provide evidence for consistent local functional activity (LCOR) decreases in individuals with autism. These alterations co-localized with in-vivo derived distributions of specific receptors and transporters covering dopaminergic, glutamatergic, GABAergic and cholinergic neurotransmission. The LCOR pattern observed in autism across both cohorts was similar to the effect of NMDA-antagonist ketamine but not to the effect of the GABAergic medication midazolam. Ketamine reduced LCOR in regions exhibiting decreases in autism. At the neurochemical level, the LCOR-neurotransmitter co-localization profile of ketamine was similar to changes observed in autism.

Previous studies investigating functional alterations in autism have often yielded inconsistent findings, potentially due to small sample sizes, the phenotypic heterogeneity of autism, or varying methodological approaches[4]. Our results support the notion of consistent group-level local activity decreases as measured using LCOR in brain regions implicated in self-referential processing, social cognition, cognitive and emotional regulation[33–36]. These findings align with recent rs-fMRI meta-analyses indicating local hypoactivity within the PCC[8], precuneus and right temporal gyri[11] and reinforce the role of functional abnormalities of the DMN in the pathophysiology of autism[10,37]. Moreover, other studies utilizing ABIDE or other datasets have reported comparable alterations in regional homogeneity (ReHo), a metric closely related to LCOR[5,6,38,39]. The central role of the DMN in the pathophysiology of autism has been further emphasized by recent efforts to identify neurobiological subtypes of autism, which have observed converging abnormalities in both the DMN and frontoparietal networks across subtypes[40,41]. Additionally, task-based fMRI studies have shown reduced activation in DMN regions during tasks requiring self-related versus other-related judgements[42,43]. Taken together, these findings suggest that decreased functional activity within DMN nodes may be related to the challenges individuals with

autism face in self-referential processing and theory of mind. Conversely, we did not observe consistent LCOR increases across both ABIDE datasets.

Most previous studies were restricted to the investigation of macroscale brain networks based on hemodynamic signals of rs-fMRI. To gain a deeper understanding of the neurobiological mechanisms underlying the emergent functional activity alterations in autism, it is crucial to integrate data from molecular neuroimaging. This approach could extend the brain connectivity framework by incorporating a molecular perspective[13,15,31]. Adopting this approach, we found LCOR alterations observed across both ABIDE datasets to be spatially related to dopaminergic, glutamatergic, GABAergic and cholinergic neurotransmission. Specifically, all co-localizations were negative with stronger LCOR reductions in autism being associated with increased availability of respective receptors and transporters in health. In line with the observed clinical heterogeneity and recently suggested existence of autism subtypes[40], these findings support the notion that multiple neurotransmitter systems may be involved in the neurobiological underpinnings of autism.

We observed the strongest similarity of autism patterns with glutamatergic NMDA and mGluR5 receptor distributions, which represents the main excitatory neurotransmitter system. Glutamate acts in a homeostatic relationship with the inhibitory GABA system balancing neuronal excitability[44]. The glutamatergic system plays an important role in brain development and neuroplasticity[14]. A balanced E/I ratio is considered to be crucial for maintaining mental health, with better overall cognitive performance in healthy children associated with a lower E/I ratio, particularly in higher-order association cortices such as the DMN nodes[45]. Conversely, an increased E/I ratio has been frequently implicated in the autism phenotype[26,27]. For instance, elevated glutamate levels, which can be found in autism in peripheral blood serum and within the central nervous system[20,46], have been shown to have neurotoxic effects potentially leading to neuronal cell death and volumetric reductions as shown in structural MRI studies[47]. A previous multimodal imaging study observed a negative association between increased glutamate concentrations in the dorsal ACC and reduced functional connectivity between the dorsal ACC and insular, limbic and parietal regions in male subjects with autism[23]. The dorsal ACC plays an important role in top-down cognitive control and adaptive, flexible behavior with decreased functional connectivity in this region linked to deficient response inhibition and more repetitive behavior in autism[33]. Consistently, higher mGluR5 densities can be observed in male adults with autism[21].

The crucial role of glutamate dysregulation in the neurobiology of autism has been highlighted by pharmacological studies manipulating the glutamatergic system[22]. Although the results are inconsistent, D-Cycloserine, a NMDA receptor agonist, showed positive effects in treatment of social difficulties and stereotypies in subjects with autism[48,49]. Conversely, the uncompetitive NMDA receptor antagonist ketamine led to reduced mentalizing performance in a social cognition task associated with increased neural activity in the superior temporal sulcus and anterior precuneus as well as increased psychotic symptoms in healthy adults[30]. In accordance, early postnatal ketamine administration in mice increased stereotyped behavior and social difficulties in later adulthood combined with elevated glutamate and reduced GABA levels in the amygdala and hippocampus[50]. Fragile X syndrome, the most commonly known single-gene cause of autism, is also linked to dysregulation of mGluR5 signaling, and pharmacological mGluR5 antagonists have demonstrated promising effects in pre-clinical and early clinical studies[51,52]. However, in clinical trials, intranasal ketamine administration in adolescents and young adults with autism did not lead to significant improvements in core autistic symptoms[53]. Conceptually, one would expect a successful treatment to normalize brain functional patterns associated with a specific clinical condition. Our own findings are therefore in line with the negative

outcome of the ketamine trial, showing that ketamine actually induces autism-like brain patterns by reducing LCOR in regions where it is also reduced in autism. Corroborating this finding, the neurochemical profile induced by ketamine significantly correlates with the neurochemical profile found in autism across all tested neurotransmitter systems. Specifically, the effect of ketamine on LCOR co-localized with the distribution of D1, NMDA and GABAa receptors suggesting that ketamine-induced manipulation of the E/I balance may lead to brain activity patterns that resemble those observed in autism. Future pharmacological studies should investigate the effectiveness of drugs manipulating this balance opposing the effects of ketamine in relieving autistic symptoms.

Optogenetic studies demonstrated that experimentally elevating the E/I ratio within the rodent medial prefrontal cortex, an important DMN hub, caused altered information processing and social divergence[54]. Notably, compensatory elevation of inhibitory neurons alleviated social divergence by balancing the E/I ratio. Potentially, an atypical expression of GABAa within the DMN contributes to autism symptom severity[24,55]. For instance, Oblak et al. (2010) found reduced GABAa receptors and benzodiazepine binding sites in the PCC and fusiform gyrus post mortem in subjects with autism[56]. Additionally, meta-analyses accumulate evidence for a negative association between local GABA concentrations and functional activation of the medial prefrontal cortex and ACC during emotion processing tasks[57]. Within our study, pharmacologically increased inhibition by GABAa-potentiator midazolam in healthy volunteers did not affect LCOR in the DMN regions where LCOR was decreased in autism. Conversely, midazolam significantly elevated LCOR in regions where LCOR was increased in autism, indicating a partial induction of autism-like brain patterns and complementing the findings observed with ketamine. At the neurochemical level, midazolam effects did not co-localize with the neurochemical signature of autism (Fig. 2B, and Supplementary Fig. S3). To date, there is modest evidence of the effectiveness of GABA modulators such as arbaclofen and acamprosate in treating autistic symptoms[29,58]. Noteworthy, distinct biological alterations, such as elevated glutamate neurotransmission or disrupted GABAergic signaling, can independently contribute to an increased E/I ratio in the brains of individuals with autism. These diverse mechanisms may ultimately converge to be associated with similar autistic phenotypes, complicating the interpretation of the underlying neurobiological pathways driving our observed results[59,60].

Several studies support the notion of altered dopamine and serotonin neurotransmission in autism, a hypothesis further reinforced by clinical observations indicating a small positive effect of the atypical antipsychotics aripiprazole and risperidone, as well as selective serotonin reuptake inhibitors, on alleviating stereotypies and aggressive behavior in some autism subgroups[14,18,61]. For instance, an over-expression of the dopaminergic system has been implicated in the modulation of stereotyped behavior in autism animal models[16]. The dopamine hypothesis of autism suggests that social difficulties may result from dysfunction in the mesocorticolimbic system, while stereotypies are thought to arise from abnormalities within the nigrostriatal circuitry[17]. We observed that LCOR reductions in autism were negatively co-localized with dopaminergic D1/D2 receptors and the dopamine transporter. These co-localizations did not yield significant correlations with specific symptom domains. Similarly, we did not find consistent in-vivo co-localization between LCOR and serotonergic neurotransmission across the ABIDE cohorts. Elevated blood serotonin levels have been reported in only a subset of autistic children (~25%), and much of the supporting evidence comes from animal models[18]. For both of these analyses, our restriction to whole-brain analyses and disregard of potential subtypes may have limited the ability to detect some regionally and subtype specific co-localization patterns.

In that regard it is also important to note that the included multi-site datasets are characterized by considerable heterogeneity with respect to the availability of demographic, medication status and clinical information (e.g., seizure history) as well as in terms of image quality. This variability may obscure more nuanced insights into the neurobiological mechanisms underlying autism. Moreover, the cross-sectional and group-averaging nature of our study design precludes any causal interpretation of our findings. Future research should aim to conduct large-scale, standardized, and longitudinal investigations to enable the identification of distinct autism subtypes and their associated neurobiological profiles.

Finally, it should be emphasized that the autistic brain may exhibit altered responses to pharmacological agents due to differences in neurotransmitter system responsivity[62]. This potential variability limits the generalizability of our pharmacological dataset, which was further constrained by the inclusion of only male participants and the impossibility to systematically account for the potential influence of prescribed medication on our findings. Furthermore, the generalizability of our findings is limited by the exclusion of autistic individuals with intellectual disability.

We find LCOR alterations in subjects with autism to negatively co-localize with the in-vivo derived distribution of dopaminergic, glutamatergic, GABAergic and cholinergic neurotransmitter systems in two large independent cohorts. This pattern of LCOR alterations in autism was similar to the effect induced by the NMDA-antagonist ketamine supporting in-vivo evidence for disturbed E/I ratio. These findings advance understanding of the pathophysiology of autism-related functional alterations and may guide new hypotheses for pharmacological interventions. Future neuro-subtyping studies with balanced gender representation aimed at disentangling the heterogeneity within autism will be crucial for identifying biologically and clinically distinct subgroups that allow for precision psychiatry approaches in autism.

## Methods
### Study design
We included data from the open-access Autism Brain Imaging Data Exchange 1 (ABIDE1) into the study, which involves 1112 participants (autism: N = 539) from 17 international centers sharing rs-fMRI and corresponding anatomical T1-weighted images as well as phenotypic data[63]. We additionally included the independently collected ABIDE2 dataset for subsequent replication of any results observed in the ABIDE1 cohort[64]. ABIDE2 includes data from 1114 participants (autism: N = 521) from 19 centers. Sex information for ABIDE participants was collected via self-report and/or clinical records; however, analyses were not stratified by sex because the primary focus of the study was on overall group effects rather than sex differences. For further information of the varying recruitment and diagnostic criteria, please see https://fcon_1000.projects.nitrc.org/indi/abide/. We excluded subjects with intellectual disability (IQ ≤ 70) because of its potential confounding effect on functional activity, subjects with missing data and subjects with excessive head motion during image acquisition (translation > 3 mm or rotation > 3°). For ABIDE2, another 3 subjects were excluded due to preprocessing failure of imaging data (Supplementary Fig. S7). Thus, 878 subjects (405 with autism) from ABIDE1 and 869 subjects (395 with autism) from the ABIDE2 dataset were included in the analyses. The mean age of participants was $17.3 \pm 8.2$ years in ABIDE1 (range 6.5–64.0 years, 15.3% female) and $15.3 \pm 9.5$ years in ABIDE2 (range 5.1–64.0 years, 22.2% female) (Table 1). All participants gave written informed consent, and all studies were approved by local ethics committees.

To investigate the relationship between molecular brain signatures and aberrant local functional activity in autism through modulation of the E/I balance in health, we analyzed data from 30 healthy male participants without a history of recreational drug use (trial registration number ACTRN12616000281493; https://www.anzctr.org.au/Trial/Registration/TrialReview.aspx?id=370230). Participants

received $15/h (pro rata) in vouchers, plus up to $60 per session for successful performance on other tasks. Reasonable travel costs were covered. Completing all three sessions earned an additional $50 bonus. This cohort was originally collected at the University of Auckland to investigate the effect of drug manipulation on EEG and rs-fMRI metrics in healthy volunteers[65]. Briefly, participants were scanned in a single-blinded, placebo-controlled, three-way crossover design, receiving ketamine, midazolam and placebo. A random-number generator was used to allocate participants into six distinct condition-order groups. The substances were administered to a sub-anesthetic level through intravenous access by an infusion pump. Racemic ketamine was given as an initial bolus of 0.25 mg/kg, followed by a continuous infusion at 0.25 mg/kg per h. Midazolam was administered similarly, with a 0.03 mg/kg bolus dose followed by a 0.03 mg/kg per h infusion. The drug infusion started 7 min into the 16-min resting-state scanning session. Sessions were spaced by a minimum of 48 h to allow for sufficient drug washout. Sex information was collected via self-report and female subjects were excluded to avoid variability in GABA levels associated with the menstrual cycle[66]. The study protocol is provided as part of the Supplementary Information. The primary outcomes of this trial have been published previously in full[65,67]. Three participants were excluded following image quality control, so that the final dataset consisted of 27 participants (M = 26.6 ± 5.9 years, 19-37 years). The study was approved by the Central Health and Disability Ethics Committee at the University of Auckland, New Zealand (Ref:[15]/CEN/254), and written informed consent was obtained from all participants.

## MRI data acquisition

As ABIDE1 and 2 data was retrospectively collected from multiple centers, acquisition of MRI images varied between sites. For individual site profiles regarding MRI protocol information, see Supplementary Table S17-18.

MRI images of the pharmacological dataset were acquired on a 3 T MRI scanner (Siemens Skyra, Erlangen, Germany) with a 20-channel head coil. For structural imaging, a 3D magnetization-prepared rapid gradient-echo (3D-MPRAGE) scan [echo time (TE) = 3.42 ms; repetition time (TR) = 2100 ms; FOV = 256 mm$^2$; flip angle 9°; 192 slices; slice thickness = 2 mm; voxel size = 1x1x1 mm] was acquired. Additionally, 246 volumes of BOLD rs-fMRI data were obtained using a T2*-weighted echo planar imaging (EPI) sequence (TE = 27 ms; TR = 2200 ms; flip angle 79°; 30 interleaved 3 mm slices; voxel size = 3 x 3 x 3 mm). During imaging, participants wore an EEG-cap, which was removed in one of the three sessions for high resolution 3D-MPRAGE structural image acquisition[65]. Participants were instructed to have their eyes open and fixated on a small cross on a projection screen.

## Preprocessing of imaging data

All rs-fMRI and structural imaging data were preprocessed using Statistical Parametric Mapping software SPM12[68] and the CONN toolbox (v22a)[69] implemented in Matlab (v2022b). Functional images were corrected for head motion and distortions (realign and unwarp), spatially normalized into MNI space and resampled to a resolution of 2 mm$^3$ isotropic. Smoothing was applied using a 6 mm full-width at half maximum (FWHM) Gaussian kernel. Mean white matter, gray matter and cerebrospinal fluid signals, as well as 24 motion parameters were regressed out before computing the voxel-based measures using CompCor, as implemented in the default CONN denoising pipeline[70]. Motion parameters were used to identify data to be excluded due to excessive head movement (translation >3 mm and rotation >3°). For the pharmaco-fMRI dataset, pre-processing was applied equivalently to the three pharmacological conditions (ketamine, midazolam and placebo). Given that both spatial smoothing and gray matter signal regression can have a significant impact on functional activity measures and subsequent

analyses[71-73], we evaluated the robustness of our primary results by excluding these preprocessing steps from the pipeline. Additionally, we removed the first four initial scans for each subject to further ensure the stability of our findings.

## Resting-state functional activity measure

To assess functional activity, we examined local synchronization (LCOR) by computing correlation at each voxel. LCOR was chosen as it provides a good approximation of local metabolism and has been recently shown to be sensitive to local pathological functional changes across different neurotransmitter systems[74,75]. LCOR is defined as the average correlation between a given voxel with other voxels in its proximity, with distances weighted by a Gaussian kernel (25 mm FWHM)[76].

## Voxel-wise group comparisons

To test for LCOR changes in autism, we performed group comparisons between subjects with autism and TD controls using the CONN toolbox. We utilized the ABIDE1 dataset for initial exploration and validated our findings using the ABIDE2 dataset. All group comparisons were corrected for age, sex, full scale IQ (FIQ), motion and site. Pairwise t-contrasts comparing autism and TD were evaluated for significance using a voxel-wise family-wise error threshold of $p < 0.05$ combined with an exact permutation-based cluster threshold (threshold free cluster enhancement (TFCE), 1000 permutations, $p < 0.05$) to control for multiple testing. The same procedure was applied to the pharmaco-fMRI dataset comparing the respective pharmacological condition with placebo condition.

## Co-localization between LCOR and neurotransmitter maps

The JuSpace toolbox was used for all further co-localization analyses[77]. Distribution of neurotransmitter systems was derived from positron emission and single photon emission computer tomography (PET, SPECT) from independent healthy volunteer populations. Neurotransmitter maps were available for serotonergic (5-HT1a, 5-HT1b, 5-HT2a, 5HT4) and dopaminergic (D1, D2) receptors, the dopamine (DAT) and serotonin transporter (SERT), the dopamine synthesis capacity (fluorodopa PET - FDOPA), GABAa, ionotropic NMDA and metabotropic glutamate receptor (mGluR5), μ-opioid (MU) and cannabinoid receptors (CB1), noradrenaline transporter (NAT) and vesicular acetylcholine transporter (VAChT).

We first examined whether autism related alterations in LCOR were spatially similar to the above neurotransmitter systems correlating the respective PET maps with the unthresholded t-contrast maps of LCOR alterations observed in autism (from autism > TD contrast). For this, we adopted a strict exploration and replication approach. We first tested for significant co-localizations in ABIDE1 applying an uncorrected p value of $p < 0.05$ (two-sided) and then tested for replication of the significant findings in ABIDE2. In addition, we used the meta-analytic Fisher's method to combine the $p$ values from both cohorts and tested for the significance of the combined $p$ values applying false-discovery Benjamini-Hochberg correction for multiple comparisons[78,79]. The default Neuromorphometrics atlas (119 regions) covering cortical, subcortical and cerebellar regions of interest was used for all analyses. We further tested for the robustness of these findings to changes in parcellation using a functional connectivity-based atlas, incorporating the Schaefer (100 parcels for cortical regions) and the Melbourne/Tian parcellations (16 parcels for subcortical regions). Partial Spearman correlations were computed adjusting for spatial autocorrelation using the gray matter probability map. Exact permutation-based p values were computed comparing the observed correlation coefficients to those obtained using permuted PET maps whilst preserving the spatial autocorrelation. For more information about the workflow of the JuSpace toolbox (v1.5), please see Dukart et al. (2021)[77].

This procedure was repeated with the pharmacological dataset testing for co-localization of the unthresholded t-contrasts for LCOR changes induced by ketamine and midazolam (both testing for increases over placebo) with the respective neurotransmitter maps. The Benjamini-Hochberg procedure was used to account for multiple comparisons[78].

### Correlation of autism-related LCOR alterations with ketamine- and midazolam-induced brain changes

To test for similarity between functional alterations induced by NMDA-antagonist ketamine and GABAa-potentiator midazolam and autism-related alterations, we first spatially correlated the respective whole-brain LCOR t-contrast maps. To assess the impact of glutamatergic and GABAergic medication on brain regions exhibiting significant differences between autism and TD groups, we separately extracted mean LCOR values per subject for ketamine, midazolam and placebo conditions from regions showing significant increases or decreases in autism in the ABIDE1 dataset. We then calculated Δ-scores for each subject in regions showing either increases or decreases in autism (separately) by subtracting LCOR values for placebo from ketamine and midazolam conditions. We then performed one-sample $t$-tests for ΔKET and ΔMDZ based on the hypothesis that both drugs induce autism-like brain patterns ($p < 0.05$, one-sided).

Lastly, we tested for similarity of the mappings of autism alterations onto all neurotransmitter systems with those observed for ketamine and midazolam. For this, we first extracted the 16 Fisher's z-transformed Spearman correlations observed for the co-localization between autism-related alterations and all of the above neurotransmitter maps. Similarly, we extracted the correlation values observed for the co-localization of changes induced by ketamine and midazolam with the same 16 neurotransmitter maps. We computed Pearson correlations between these neurochemical co-localization profiles obtained for autism, ketamine and midazolam conditions.

### Relationship of LCOR-neurotransmitter co-localizations with clinical phenotypes of autism

Next, we tested whether the significant co-localizations of LCOR with specific neurotransmitters across both ABIDE cohorts relate to symptom severity in autism. For this, we calculated individual z-scores in each region for each subject with autism relative to the mean of the respective TD control group and correlated the obtained z-maps with the respective neurotransmitter maps. The obtained individual Fisher's z-transformed coefficients, indicating the strength of LCOR-neurotransmitter co-localization on individual level, were used to perform Pearson correlations with clinical measures of autistic symptom severity within IBM SPSS Statistics (Version 27)[80].

We used the Autism Diagnostic Observation Schedule (ADOS) as a measure of autism symptom severity[81]. The ADOS is a semi structured, standardized measure of autistic phenotypes which consists of four age-adjusted 30-min modules according to the level of expressive language covering the following three subdomains: communication difficulties, reciprocal social interaction (RSI), and stereotyped behaviors and restricted interest (SBRI) (Table 1). We correlated the symptom severity of each subdomain with the individual Fisher's z-transformed coefficients to explore how variations in neurotransmitter systems may be linked to distinct symptom profiles in autism. ADOS data were available for 300 participants with autism in the ABIDE1 cohort and 238 participants with autism in the ABIDE2 cohort (Table 1).

### Reporting summary

Further information on research design is available in the Nature Portfolio Reporting Summary linked to this article.

### Data availability

The pharmacological data generated in this study are available under restricted access due to ethical approval and informed consent restrictions; access can be obtained by contacting the corresponding author (Prof. Juergen Dukart, j.dukart@fz-juelich.de) and providing a signed data use agreement and institutional ethics committee approval. Raw pharmacological data are protected and are not available due to data privacy laws. The processed pharmacological data are available in the Source Data files accompanying this paper and contain only aggregated summary values that do not permit identification of individual participants. The autism neuroimaging data used in this study are available in the Autism Brain Imaging Data Exchange (ABIDE) repository (https://fcon_1000.projects.nitrc.org/indi/abide/; https://doi.org/10.1038/mp.2013.78). The original ABIDE data are de-identified but subject to the repository's data use terms, and cannot be redistributed directly. The processed ABIDE data generated in this study are available in the Source Data files accompanying this paper and contain only aggregated and derived metrics that do not permit identification of individual participants. Source data, results from whole-brain voxel-wise analyses, and thresholded result maps have been deposited in Zenodo (https://doi.org/10.5281/zenodo.16904492). Source data are provided with this paper.

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

## Acknowledgements

We thank all participants for taking part in this research. J.D. has received funding from the European Union's Horizon 2020 research and innovation program under grant agreement No. 826421, "TheVirtualBrain-Cloud." S.B.E. has received funding from the European Union's Horizon 2020 research and innovation program under grant agreement No. 945539 (Human Brain Project SGA3). L.S. has received funding from the Forschungskommission of the Medical Faculty at the Heinrich-Heine-University Düsseldorf. ABIDE I is supported by NIMH (K23MH087770), NIMH (R03MH096321), Leon Levy Foundation, Joseph P. Healy, and the Stavros Niarchos Foundation. ABIDE II is supported by NIMH (5R21MH107045), NIMH (5R21MH107045), Nathan S. Kline Institute of Psychiatric Research, Joseph P. Healey, Phyllis Green, and Randolph Cowen. No funding was received for this work.

## Author contributions

Conceptualization: P.G., J.D. Methodology: P.G., J.D. Formal analysis: P.G., J.D. Investigation: P.G., J.D. Data curation: J.F.H., A.F., S.M., J.D. Visualization: P.G., J.K., J.D. Project administration: J.D. Supervision: J.D. Writing–original draft: P.G. Writing–review & editing: J.K., J.F.H., A.F., S.L.V., S.M., S.B.E., L.S., J.D.

## Funding

## Competing interests

The clinical trial was funded by F. H.–La Roche Ltd. J.F.H. is a current employee of F. Hoffmann–La Roche Ltd. and received support in the form of salaries. This employment has not influenced the interpretation or conclusions of this research. A.F. has been a paid independent consultant for Viridia Life Sciences and MindBio Therapeutics. This relationship did not exist during the time of data collection and has not influenced the design, analysis, interpretation or conclusion of this research. The remaining authors declare no competing financial or non-financial interests.

## Additional information

[1]Institute of Neurosciences and Medicine, Brain & Behaviour (INM-7), Research Centre Juelich; Wilhelm-Johnen-Straße 1, Juelich, Germany. [2]Department of Psychiatry and Psychotherapy, Medical Faculty and University Hospital Duesseldorf, Heinrich Heine University Duesseldorf; Bergische Landstraße 2, Duesseldorf, Germany. [3]Institute of Systems Neuroscience, Medical Faculty and University Hospital Duesseldorf, Heinrich Heine University Duesseldorf; Moorenstraße 5, Duesseldorf, Germany. [4]Roche Pharma Research and Early Development, Neuroscience and Rare Diseases, Roche Innovation Center Basel, F. Hoffmann–La Roche Ltd., Basel, Switzerland. [5]School of Pharmacy, Faculty of Medical and Health Sciences, University of Auckland; 85 Park Road, Grafton, Auckland, New Zealand. [6]Max Planck School of Cognition; Stephanstraße 1A, Leipzig, Germany. [7]Max Planck Institute for Human Cognitive and Brain Sciences; Stephanstraße 1A, Leipzig, Germany. [8]Department of General Psychiatry 2, LVR-Klinikum Duesseldorf, Heinrich Heine University Duesseldorf; Bergische Landstraße 2, Duesseldorf, Germany. [9]Department of Psychiatry and Psychotherapy, University Hospital, Ludwig Maximilians University Munich; Nußbaumstraße 7, München, Germany. ✉e-mail: j.dukart@fz-juelich.de

