## [Transparent Peer Review file · Nature Communications]

Local activity alterations in individuals with autism correlate with neurotransmitter properties and ketamine-induced brain changes

Corresponding Author: Professor Juergen Dukart

Version 0:

Reviewer comments:

Reviewer #1

(Remarks to the Author)

This article examined differences in a local connectivity measure (LCOR) between individuals with autism and typical individuals from the ABIDE database. The results show that individuals with ASD show increased LCOR in temporal areas and decreased LCOR related to the DMN. Group differences in LCOR were then correlated with neurotransmitter maps derived from healthy subjects showing associations between LCOR differences and neurotransmitter representation within the brain.

While the study investigates an important aspect of ASD neurobiology, there are several concerns that dampen my enthusiasm for the article. The largest is the lack of multiple correction comparison for most statistical tests and the description of the comparison between the LCOR maps and neurotransmitter maps (the main findings of the article) is not very clear. Further comments are listed below.

The LCOR measure used is essentially akin to the regional homogeneity (ReHo) approach that has been well documented in ASD using ABIDE data (e.g., Dajani & Uddin, 2016; Ren et al., 2023) and other data (e.g., Zhao et al., 2022) yet there is little mention of previous research in the introduction. The introduction makes a vague mention of some previous connectivity results but nothing substantial despite the fact that there are a number of studies that have examined local connectivity in ASD.

The use of smoothing is usually not recommended for local connectivity measures such as ReHo as smoothing has large effects on such metrics. How do different smoothing kernels (or no smoothing) influence the results?

I assume that average "gray matter" regression was global signal regression? Due to the known influences of global signal regression on functional connectivity and group differences, the results should be replicated without it to show the effects are solely dependent on global signal regression.

Although site was controlled for, there are better methods available such as CompCorr to correct for site differences that are tailored to this specific situation.

The comparison of the LCOR alterations with the neurotransmitter maps is a little unclear. Although I can see the value in exploratory ROI analyses, why not conduct an overall analysis identifying areas that are significantly different in LCOR between TD and ASD that overlap with the areas that significantly differentiate between drug administration conditions?

Everything else would seem to be comparing areas of the brain that are not related to group diagnosis (e.g., differences in LCOR) or drug conditions (e.g., ketamine) but are merely random fluctuations. That is, while these associations may mean these measures are related to each other, the measures themselves are not related to group differences or drug administration themselves making any interpretation of their relationship unclear with regards to the main aim of the study.

Were the neurotransmitter maps unthresholded also?

The neuromorphometrics atlas seems to be an anatomical atlas? It is probably better to use a functional atlas given that you are looking at a measure of functional connectivity as there is large differentiation between anatomical and functional parcellations.

Reviewer #2

(Remarks to the Author)

This is a very interesting paper using combined virtual functional histology merged with information from different drugs to explore whether functional differences in autism are associated with specific neurotransmitter systems which were subsequently tested. While this is an interesting paper, I do think the paper has some significant issues.

First of all, the language is extremely pathologising. Recognising recent papers in both the US and UK (e.g. Bottema-beutel and colleagues) about preferred language, as well as recent studies showing the benefit of neuro-affirmative language in reducing stigma (compared to a deficit heavy language as used here), are really critical here. While I can accept that authors may prefer to use "DSM-5" language, they are othering neurodivergent individuals (incl. myself) in their language and should, at the least, address this in the intro or a footnote. This is quite simple, using Autism, instead of ASD. One approach could be "We are working on Autism Spectrum Disorder, henceforth referred to as "Autism"". Similarly, "participants" is much nicer language than the old school "subjects", and try to avoid "deficits", "dysfunction" and the like. Ultimately, this work is done FOR the community, so using community preferred language could be considered.

Secondly, the big issue here is that the autistic brain responds different to drugs (as evidenced by many single drug probe studies, and recently shown in this preprint too: <https://www.medrxiv.org/content/10.1101/2024.11.08.24316969v1>) AND the "many ways lead to Rome" notion that different biological differences in autism lead to similar behavioural phenotypes. In other words, person A might have an affected GABA transporter, another an NMDA receptor, which both lead to a similar imbalance in "E/I" (a non specific umbrella term) and behaviour. This heterogeneity isn't captured, explored, or explained. The samples are not generalisable and the pharmacological group is very small and only consists of healthy male adults which significantly affects generalisability and is a weakness of this study.

P4; It's not immediately relevant how rs-fMRI can support diagnosis? MRI's are expensive and there currently are no biomarkers at all for autism and most studies are mere case control. Autism is defined behaviourally. I don't really think the authors are right in this statement and they should consider removing it. I would argue rs-fMRI can aid in understanding the nature of behavioural manifestations as part of the autistic phenotype. Beyond that the authors actually mostly look at case control differences, and they do not address heterogeneity well. This also refers to the discussion (p17) discussing different outcomes of clinical trials

I would also disagree that the focus to date has been on glutamate. The focus has been on the E/I imbalance theory (which should be mentioned first) and in fact, predominantly on GABA.

This sentence comes out of nowhere: "It also remains uncertain whether pharmacologically induced E/I modulation as elicited by glutamatergic medication ketamine and GABAergic medication midazolam in healthy subjects results in macroscale functional reorganization akin to that observed in ASD^{27–29}." and generally I find the introduction poorly written and justified.

Figure 1 and 2 could do with stats on the plot for interpretability.

In the style of writing for Nature comms, I think the results could do with a bit more context on the samples for the two drugs and as above, motivation, details as this comes out of nowhere.

This statement does not seem supported by data at all: "This suggests that decreased LCOR in DMN nodes may relate to difficulties of individuals with ASD in self-referential processing and inferring mental states of others ("theory of mind"). Our findings further support recent efforts to identify neuro-subtypes in ASD, which have observed converging abnormalities in both the DMN and frontoparietal networks across subgroups^{41,42}." and the discussion is generally poorly written, reading more like a lit summary rather than contextualising the findings.

This is making the findings limited and not generalisable: "We excluded subjects with intellectual disability (IQ \leq 70) to reduce variability associated with low-functioning ASD" "Low functioning" is not a preferred term, this is absolutely a spectrum (see recent discussions surrounding profound autism). I would argue this is not about ASD but simply about inclusion and exclusion of intellectual disability. But it limits generalisability.

The drug datasets are extremely small and only consist of healthy males which again, limits generalisability, does not inform on autism and I would say is a significant downside of this study. It doesn't take into account gender effects at all and as such is an outdated paradigm.

It's not clear how the ADOS was used.

Reviewer #3

(Remarks to the Author)

Summary: This article presents a sophisticated and novel approach to the E/I imbalance theory of autism. Researchers have capitalized on the large ABIDE I and II datasets, conducted a smaller scale medication trial of two agents that manipulate Glutamate and GABA systems, and compared spatial locations of hyper and hypo activation in autism to the pharmacological study results and to molecular imaging data. The findings are likely to have an impact on the field, spurring more such pharmacological trials for comparison to large databases. I have a few comments to improve the manuscript:

1. The phrasing of the sentence on page 5, last paragraph, "...show consistent local hyper- and hypoactivity in sensorimotor and DMN areas..." can be read in at least two ways with different meanings. Either it reads that the sensorimotor area is hyperactive while the DMN is hypoactive (likely intended meaning), or that each area is both hyperactive and hypoactive. The later meaning doesn't really make sense, so I suggest re-writing this to emphasize which area(s) exhibit which findings.
2. Typically, one discards the first 8 seconds or so of EPI fMRI data (sometimes referred to as "dummy" scans) so that the remaining scans represent a steady-state. I do not see mention of this. The authors should clarify if this was done, or provide a defense of not implementing it if not done.
3. Page 25, top sentence: provide citation for Benjamini-Hochberg FDR procedure.
4. If drug effects were expected to produce changes in both directions (ASD>TD or TD>ASD) on calculated delta scores, why were one-sided tests conducted? I'm not sure this is justifiable.
5. In the supplementary figures S2 and S3, the threshold used for visualization of the voxel-wise results should be stated in the figure notes.
6. In the discussion of mGluR5, it would be worth referencing the interest in this as a treatment target in Fragile X Syndrome, since that disorder is one of a few single gene disorders associated with autism.

Reviewer #4

(Remarks to the Author)

This study looks at how brain activity differs in people with autism spectrum disorder (ASD) and what might be causing those differences. Researchers analyzed brain activity at rest in people with ASD and compared it to people without ASD using data from two large groups. They found that people with ASD had consistently lower brain activity in certain areas, especially in parts of the brain involved in thinking about oneself and others (the default mode network).

These changes in brain activity were linked to differences in several brain chemical systems, including those involving dopamine, glutamate, GABA, and acetylcholine—essential chemicals that help brain cells communicate. The researchers also tested how certain drugs affect brain activity. They found that ketamine, which affects glutamate, caused brain changes similar to those seen in ASD, while midazolam, which affects GABA, did not.

This suggests that disruptions in the balance of brain chemicals might play a role in the brain differences seen in ASD, helping us better understand its underlying causes and offering candidate biomarkers to inform treatment development. Strengths of the work include:

1. Using the ABIDE datasets provides for easy replication of the analysis and extension of the work by other researchers.
2. The test-retest approach in ABIDE 1 vs. ABIDE 2 is very nice.
3. The impact of the findings is high. The work will likely lead to new avenues for treatment development in ASD.
4. While focused on autism, the work has broad implications for our understanding of brain function and neurological conditions broadly.
5. The use of a placebo control, as well as a comparison of two different drugs targeting two different receptor systems, in PET imaging is compelling.

Minor weaknesses are:

1. The authors confused me a bit in setting the stage in the introduction. I suspect this was because of the need to be concise for this journal's format. Specifically, I know why it is fascinating that the changes in ASD may mirror those induced by ketamine and/or midazolam, but presently, this isn't well articulated in the introduction. This clarification will help a broader audience and ASD-focused audience appreciate the importance of the work. To keep the manuscript concise, you could take a bit from the excellent discussion of this point from the discussion section and move it here or duplicate in both places to ensure the reader follows the motivation clearly before getting into the results.
2. Please see the questions below. Depending on your responses, some of these questions could be points that you need to consider as potential limitations to be discussed in contextualizing your findings.

Questions for the authors:

1. To what degree are subjects independent across ABIDE 1 and 2? Are some subjects, likely at different points in development, duplicated in each collection?
2. What information is available regarding medications taken by subjects in the ABIDE collections? I ask because GABA-, Dopamine-, and Serotonin-targeting medications are commonly prescribed for associated conditions in autistic patients.
3. Related to the question above, is information available about the presence of seizures for the ABIDE collections?

4. In line 94, should "neurotransmitter" be plural here?

5. Are the Social Responsiveness Scale scores for participants available in ABIDE? If so, this might be a more powerful tool to examine behavioral correlates of the observed activation patterns.

In sum, this is an exciting, impactful study. The paper is well-written and compelling. The conclusions closely follow from the results. The work will be of interest to a broad audience.

I sign all of my reviews. – Kevin Pelphrey

Version 1:

Reviewer comments:

Reviewer #1

(Remarks to the Author)

I appreciate the author's responses to my previous comments which they have addressed. However, this comment does not really address my concern which is that the interpretation remains unclear of how this result speaks to the relationship between ASD and drug related effects. As mentioned in my original comment, two things can be correlated even if they are not related.

If I correlated subscales of the ADOS with subscales of some measure of depression and found they were related, this relationship would be questionable if neither the total ADOS nor the total depression scale passed the threshold for a clinical diagnosis. I am having a hard time seeing how one should interpret the relationship between the two as there is no official diagnosis of either ASD nor depression even though they are significantly correlated. It is the same in this situation.

Response: We apologize again for the unclear description of the applied methodology. The reviewer is absolutely correct that regions which do not display any autism (or drug related) effects would show random fluctuations in their response. However, first of all, those random fluctuations are in the first place random and therefore not expected to induce any significant correlations exceeding those observed in randomly generated maps with the same spatial smoothness. If only such random effects prevail the statistics would simply become not significant.

Correlation does not equal causation and just because two things are associated, it doesn't mean they are related as claimed, nor does it mean that the interpretation of their relationship is clear.

"Second, the reason of using t-maps (as a measure of effect size) in our study and other similar spatial co-localization studies is exactly to minimize the contribution of this type of random fluctuations as those (considering the large sample size) will only generate weak effect sizes tending close to zero whilst mostly only true positive effects will induce larger positive and negative t-values and therefore contribute to the actual spatial correlations that deviate from a random distribution of such correlations."

The sample size is small for the drug effects causing concern of spuriously large associations.

"Whilst it is conceptually possible to restrict the spatial co-localization approach to only regions which for example have significant effects in autism, this type of restriction creates other types of issues which completely outweigh its advantages. For example, correlation of only a preselection of autism-related regions with neurotransmitter maps may artificially narrow down the actual range of the neurotransmitter information, i.e. regions with only very low or very high expression of a specific receptor get selected. In the case of the low expression, the correlation in those regions will then appear meaningless as all selected regions basically do contain any meaningful receptor signal and the variance therefore mostly reflects noise. In the case of high expression, a true correlation may be lost as we only look at regions with similarly high expression and completely lose the contrast against regions which have low or moderate expression of the respective signal. As we have no valid ways to prevent or detect such thresholding induced biases without looking at the actual unthresholded maps, the direct use of unthresholded maps combined with proper non-parametric statistics appears more appropriate and rigorous."

While this approach offers more possibilities to find significant effects, that is not really a justification for forcing an interpretation on the results that is not supported. A region should be selected because it is related to ASD and related to drug administration, not because selecting more regions offers more chances to find a result.

As in the above example I gave, that approach would necessitate that I examine only those with a diagnosis of ASD and a diagnosis of depression, while this approach that is offered would examine all subjects results regardless of diagnosis. The former comparison is easily justified as looking at the relationship between ASD and depression while for the latter it is harder to state exactly what the relationship is.

I am unsure how one can state what the relationship is between the two variables used in this study in this particular way, but it doesn't seem like one can claim it is a relationship strictly between ASD and drug administration.

Reviewer #2

(Remarks to the Author)

To the authors, I want to commend the authors on a thorough well thought out response the reviewer comments. All the comments I made previously have been addressed appropriately and I appreciate the authors taking the time to significantly revise the manuscript. While I have some minor issues that cannot be addressed (e.g. dealing with mild ID, only males, and conceptualisation of the ADOS as an outcome) I don't believe any further discussion/revisions will benefit the manuscript. I am happy with this.

Reviewer #4

(Remarks to the Author)

I am pleased with the revisions and answers to my questions. The paper is now in great shape.

Version 2:

Reviewer comments:

Reviewer #1

(Remarks to the Author)

The authors have addressed all of my concerns

We thank the reviewers for providing their positive reviews of our manuscript. We have thoroughly revised and complied with the reviewers' suggestions, and we believe that the paper has substantially benefited from this review. Authors' point-by-point responses to the reviewers' comments are provided below and corresponding changes in the revised manuscript are highlighted in yellow color.

Reviewer: 1

“This article examined differences in a local connectivity measure (LCOR) between individuals with autism and typical individuals from the ABIDE database. The results show that individuals with ASD show increased LCOR in temporal areas and decreased LCOR related to the DMN. Group differences in LCOR were then correlated with neurotransmitter maps derived from healthy subjects showing associations between LCOR differences and neurotransmitter representation within the brain.

***Response:** We thank the reviewer for this positive feedback on our manuscript. We have now thoroughly revised the manuscript complying with the reviewer's suggestions.*

While the study investigates an important aspect of ASD neurobiology, there are several concerns that dampen my enthusiasm for the article. The largest is the lack of multiple correction comparison for most statistical tests and the description of the comparison between the LCOR maps and neurotransmitter maps (the main findings of the article) is not very clear. Further comments are listed below.”

***Response:** We thank the reviewer for raising the important issue of multiple comparisons correction and apologize if this was not sufficiently explained in our manuscript. More specifically, a multiple comparison correction serves the only purpose to control the number of false positive findings when conducting multiple statistical tests. As in our case, we had two datasets covering the same question, we adopted an exploration and replication approach to achieve the same control for false positives which is often considered the more conservative approach. More specifically, we first tested for significant colocalization effects in the ABIDE1 cohort resulting in 10 out of 16 tested associations being significant. And then for significant associations in the ABIDE2 replicating 7 out of this 10 associations in the ABIDE2 dataset. This step-wise procedure typically results in an even more conservative correction for multiple comparisons. To illustrate this issue, we have adopted a common meta-analytic approach of combining p-values from both cohorts using the conservative Fisher's method and applied a false-discovery rate correction to the resulting p-values. This resulted in 10 out of 16 associations surviving the correction for multiple comparisons including all of the 7 associations that were replicated in the ABIDE 2 cohort. In addition, the associations with 5-HT1b, CBI and NAT would be considered significant in this meta-analytic approach despite not being significant in one of the two datasets. We have now added this analysis to our manuscript (p.8):*

“The meta-analytically combined p -values from both cohorts survived the correction for multiple comparisons for all of these replicated co-localizations (see Supplementary Table S9). In addition, the combined p -values for the associations with 5-HT1b, CB1 and NAT were also significant after correction for multiple comparisons.”

Additionally, we explained this approach in more detail in the respective methods section (p.25-26):

“In addition, we used the meta-analytic Fisher’s method to combine the p -values from both cohorts and tested for the significance of the combined p -values applying false-discovery Benjamini-Hochberg correction for multiple comparisons (Benjamini & Hochberg, 1995; Fisher, 1934).”

However, to remain on the conservative side we kept the restriction of the discussion to only those effects that replicated in both cohorts.

	p -value (ABIDE1)	p -value (ABIDE2)	p -value (meta-analytically)
5HT1a	0.1349	0.7213	0.3240
5HT1b	0.0809	< 0.001***	0.0008***
5HT2a	0.6993	0.998	0.9489
5HT4	0.024*	0.7423	0.0896
SERT	0.6553	0.7982	0.8620
D1	0.005**	0.002**	0.0001***
D2	< 0.001***	< 0.001***	0.0000***
DAT	0.033*	0.028*	0.0074**
FDOPA	0.034*	0.3996	0.0720
NMDA	0.009**	0.009**	0.0008***
mGluR5	0.004**	0.003**	0.0001***
GABAa	0.005**	0.035*	0.0017**
CB1	0.3816	0.011*	0.0272*
MU	0.4905	0.34	0.4655
NAT	0.024*	0.0639	0.0115*
VACHT	0.003**	0.006**	0.0002***

Supplementary Table S9. Meta-analytically combined p -values from co-localization results in both ABIDE cohorts using Fisher’s method. ABIDE = Autism Brain Imaging Data Exchange; * = $p < .05$; ** = $p < .01$; *** = $p < .001$.

“The LCOR measure used is essentially akin to the regional homogeneity (ReHo) approach that has been well documented in ASD using ABIDE data (e.g., Dajani & Uddin, 2016; Ren et al., 2023) and other data (e.g., Zhao et al., 2022) yet there is little mention of previous research in the introduction. The introduction makes a vague mention of some previous connectivity results but nothing substantial despite the fact that there are a number of studies that have examined local connectivity in ASD.”

***Response:** We sincerely thank the reviewer for bringing this important point to our attention. Indeed, several studies have investigated ReHo in ABIDE data. We had previously mentioned this in our discussion on page 15, stating: “Other studies utilizing ABIDE or other datasets have reported comparable alterations in regional homogeneity (ReHo), a metric closely related to LCOR.”*

To address the reviewer’s suggestion more thoroughly and to strengthen our contextualization of prior research, we have now incorporated references to more recent and well-powered studies, including Ren et al. (2023)¹ and Zhao et al. (2022)², both of which reported findings consistent with ours. These additions have been made in both the introduction (p. 4) and discussion (p. 15) sections to better situate our work within the broader literature.

“The use of smoothing is usually not recommended for local connectivity measures such as ReHo as smoothing has large effects on such metrics. How do different smoothing kernels (or no smoothing) influence the results?”

I assume that average “gray matter” regression was global signal regression? Due to the known influences of global signal regression on functional connectivity and group differences, the results should be replicated without it to show the effects are solely dependent on global signal regression.”

***Response:** We thank the reviewer for these suggestions. In response, we repeated the preprocessing of functional activity for both ABIDE1 and ABIDE2 datasets, this time omitting both spatial smoothing and gray matter global signal regression during the denoising step. Additionally, we removed the first 4 initial scans of every nifti-file so that the remaining scans represent a steady-state (see suggestion from reviewer #3). Subsequently, we re-evaluated the group differences in LCOR between autistic individuals and TD controls, as well as the alterations in neurotransmitter co-localization profiles.*

For ABIDE1, the LCOR group differences exhibited a similar pattern to our original findings, particularly the reduced LCOR in autism, though with a spatially reduced extent (as expected due to the absence of smoothing; see Supplementary Figure S1). All replicated LCOR-neurotransmitter co-localizations remained statistically significant despite the preprocessing changes (Supplementary Table S6). Notably, the co-localization with mGluR5, NMDA, and GABA_A receptors demonstrated even stronger effects in this revised analysis.

For ABIDE2, we observed a comparable but spatially reduced LCOR alteration pattern between autism and TD controls (Supplementary Figure S2). Regarding neurotransmitter co-

localization, associations with serotonergic 5HT1b, dopaminergic D1 and D2, mGluR5, GABA_A, and CB1 receptors remained significant (Supplementary Table S7). The co-localization with NMDA receptors was now marginally significant, whereas associations with mGluR5 and GABA_A receptors showed even stronger effects.

These findings indicate that all of our primary results remained robust and statistically significant even after changes to the pre-processing pipeline such as the exclusion of smoothing and omission of gray matter signal regression. We have added these results to the results section and referred to the respective section in the supplements for details (p.7; p.8):

“The voxel-wise findings were robust but spatially less extended following changes in preprocessing pipeline including removing the first four initial scans, smoothing and gray matter signal regression (Supplementary Figures S1-2).”

“The co-localization profiles were robust to changes in pre-processing pipeline including removing the first four initial scans and omitting smoothing and gray matter signal regression (see Supplementary Tables S6-7).”

Additionally, we explained this procedure in the methods section (p.24):

“Given that both spatial smoothing and gray matter signal regression can have a significant impact on functional activity measures and subsequent analyses (Herrera et al., 2016; Triana et al., 2020; Vos de Wael et al., 2017), we evaluated the robustness of our primary results by excluding these preprocessing steps from the pipeline. Additionally, we removed the first four initial scans for each subject to further ensure the stability of our findings.”³⁻⁵

Supplementary Figure S1. Results of voxel-wise comparisons between autism and TD controls in the ABIDE1 dataset. Left: Original results from the manuscript. Right: Results after removing first four initial scans, smoothing and gray matter signal regression from preprocessing pipeline.

Co-localization	ABIDE1 (original results)		ABIDE1 (without smoothing and GM regression)	
	Spearman r	p -value	Spearman r	p -value
5HT1a	.1982	.1349	.0939	.4785
5HT1b	-.2480	.0809	-.2707	.0499*
5HT2a	.0555	.6993	-.0714	.5984
5HT4	.1945	.0240*	.0679	.4665
SERT	-.0393	.6553	-.1304	.1549
D1	-.2634	.0050**	-.2831	.0030**
D2	-.3216	< .0010***	-.3139	< .0010***
DAT	-.1867	.0330*	-.1864	.0470*
FDOPA	-.2000	.0340*	-.1733	.0509
NMDA	-.3318	.0090**	-.3781	.0020**
mGluR5	-.3786	.0040**	-.4149	.0020**
GABA _A	-.2478	.0050**	-.2777	.0020**
CB1	-.1307	.3816	-.1675	.2747
MU	-.1056	.4905	-.0864	.5824
NAT	-.2018	.0240*	-.1663	.0709
VACHT	-.2409	.0030**	-.1893	.0410*

Supplementary Table S6. Statistical data of co-localizations between LCOR alterations in autism compared to TD and neurotransmitter systems in ABIDE1 after removing the first four initial scans, smoothing and gray matter signal regression. ABIDE = autism brain imaging data exchange; GM = gray matter; 5HT = 5-hydroxytryptamine (serotonin receptor); SERT = serotonin transporter; D1/D2 = dopamine receptor; DAT = dopamine transporter; FDOPA = fluorodopa; NMDA = N-Methyl-D-Aspartat receptor; mGluR5 = metabotropic glutamate receptor; GABA_A = γ -aminobutyric acid type A receptor; CB1 = cannabinoid receptor; MU = μ -opioid receptor; NAT = noradrenaline transporter; VACHT = vesicular acetylcholine transporter; * = $p < .05$; ** = $p < .01$; *** = $p < .001$.

Supplementary Figure S2. Results of voxel-wise comparisons between autism and TD controls in the ABIDE2 dataset. Left: Original results from the manuscript. Right: Results after removing first four initial scans, smoothing and gray matter signal regression from preprocessing pipeline.

Co-localization	ABIDE2 (original results)		ABIDE2 (without smoothing and GM regression)	
	Spearman r	p -value	Spearman r	p -value
5HT1a	.0492	.7213	-.0806	.5165
5HT1b	-.4308	< .0010***	-.4054	.0020**
5HT2a	-.0002	.9980	-.1188	.3456
5HT4	-.0334	.7423	-.0195	.7962
SERT	-.0245	.7982	.0162	.7962
D1	-.2778	.0020**	-.2787	.0030**
D2	-.3239	< .0010***	-.2385	.0070**
DAT	-.1963	.0280*	-.1693	.0640
FDOPA	-.0794	.3996	-.0236	.8012
NMDA	-.3109	.0090**	-.2125	.0689
mGluR5	-.3923	.0030**	-.4075	< .0010***
GABAa	-.1864	.0350*	-.3133	< .0010***
CB1	-.3348	.0110*	-.3192	.0110*
MU	-.1484	.3400	-.0671	.6314
NAT	-.1641	.0639	-.1744	.0569
VACHT	-.2378	.0060**	-.0907	.3237

Supplementary Table S7. Statistical data of co-localizations between LCOR alterations in autism compared to TD and neurotransmitter systems in ABIDE2 after removing the first four initial scans, smoothing and gray matter signal regression. ABIDE = autism brain imaging data exchange; GM = gray matter; 5HT = 5-hydroxytryptamine (serotonin receptor); SERT = serotonin transporter; D1/D2 = dopamine receptor; DAT = dopamine transporter; FDOPA = fluorodopa; NMDA = N-Methyl-D-Aspartat receptor; mGluR5 = metabotropic glutamate receptor; GABAa = γ -aminobutyric acid type A receptor; CB1 = cannabinoid receptor; MU = μ -opioid receptor; NAT = noradrenaline transporter; VACHT = vesicular acetylcholine transporter; * = $p < .05$; ** = $p < .01$; *** = $p < .001$.

“Although the site was controlled for, there are better methods available such as CompCorr to correct for site differences that are tailed to this specific situation.”

***Response:** We sincerely appreciate the reviewer’s suggestion regarding the use of CompCor to correct for site-related differences. We apologize for not having described the processing in sufficient detail in the initial version of our manuscript.*

CompCor is already part of the default denoising pipeline in CONN, which we employed in our preprocessing steps. We have now added this detail to the manuscript (p.24):

“Mean white matter, gray matter and cerebrospinal fluid signals, as well as 24 motion parameters were regressed out before computing the voxel-based measures using CompCor, as implemented in the default CONN denoising pipeline.”

Importantly, while CompCor effectively removes confounds related to physiological noise, it is not specifically designed to correct for site differences, which encompass a broader range of factors such as scanner hardware, acquisition protocols, and participant demographics. Therefore, we opted to also keep the part for an explicit statistical control approach that directly models and accounts for these variations in addition to denoising performed by CompCor.

“The comparison of the LCOR alterations with the neurotransmitter maps is a little unclear. Although I can see the value in exploratory ROI analyses, why not conduct an overall analysis identifying areas that are significantly different in LCOR between TD and ASD that overlap with the areas that significantly differentiate between drug administration conditions?”

***Response:** We thank the reviewer for raising this important issue of the reason for using the hierarchical approach in our study and apologize if the purpose of our analysis workflow was not sufficiently clear. More specifically, when designing our study we have specifically chosen the hierarchical approach of first testing for differences between autism and TD groups and then using the resulting T-contrasts for correlation with the neurotransmitter maps as well of the testing for the effects of both drugs on the exact regions that we have found LCOR to be significantly different between autism and TD. As our whole objective is built on the idea of testing for evidence of excitation and inhibition imbalance in autism, in our opinion, the chosen hierarchical approach is more rigorous in addressing this objective and less exploratory than conducting conjunction analysis as suggested by the reviewer. More specifically, the issue of a conjunction analysis, as suggested by the reviewer, is that even if we would have found some significant overlap between drug and autism-related effects in example in several voxels or a set of regions, this would only have provided evidence that both affect to some extent overlapping circuitries but not that the actual circuitry underlying both mechanisms is anyhow similar. In contrast, adopting a hierarchical approach and using the regional mask derived from autism individuals allows us to test if and how the functional network altered by ketamine is similar to the one observed in autism and (using the mask of significant findings) if regions*

that are found to be altered in autism individuals are modulated by ketamine and midazolam. Combining both types of analyses provides compelling evidence that the functional alteration patterns induced by ketamine are indeed similar to the overall effect observed on group-level in the autistic population.

“Everything else would seem to be comparing areas of the brain that are not related to group diagnosis (e.g., differences in LCOR) or drug conditions (e.g., ketamine) but are merely random fluctuations. That is, while these associations may mean these measures are related to each other, the measures themselves are not related to group differences or drug administration themselves making any interpretation of their relationship unclear with regards to the main aim of the study.”

Response: *We apologize again for the unclear description of the applied methodology. The reviewer is absolutely correct that regions which do not display any autism (or drug related) effects would show random fluctuations in their response. However, first of all, those random fluctuations are in the first place random and therefore not expected to induce any significant correlations exceeding those observed in randomly generated maps with the same spatial smoothness. If only such random effects prevail the statistics would simply become not significant. Second, the reason of using t-maps (as a measure of effect size) in our study and other similar spatial co-localization studies is exactly to minimize the contribution of this type of random fluctuations as those (considering the large sample size) will only generate weak effect sizes tending close to zero whilst mostly only true positive effects will induce larger positive and negative t-values and therefore contribute to the actual spatial correlations that deviate from a random distribution of such correlations. Whilst it is conceptually possible to restrict the spatial co-localization approach to only regions which for example have significant effects in autism, this type of restriction creates other types of issues which completely outweigh its advantages. For example, correlation of only a preselection of autism-related regions with neurotransmitter maps may artificially narrow down the actual range of the neurotransmitter information, i.e. regions with only very low or very high expression of a specific receptor get selected. In the case of the low expression, the correlation in those regions will then appear meaningless as all selected regions basically do contain any meaningful receptor signal and the variance therefore mostly reflects noise. In the case of high expression, a true correlation may be lost as we only look at regions with similarly high expression and completely lose the contrast against regions which have low or moderate expression of the respective signal. As we have no valid ways to prevent or detect such thresholding induced biases without looking at the actual unthresholded maps, the direct use of unthresholded maps combined with proper non-parametric statistics appears more appropriate and rigorous.*

“Were the neurotransmitter maps unthresholded also?”

Response: *Yes, the neurotransmitter maps are also unthresholded for the very same argument as provided to the previous comment of the reviewer. We hope that with the above reasons we could convince the reviewer that the use of such unthresholded maps is the best viable alternative to avoid incorrect conclusions of the potential associations between both types of signals due to artificially narrowed ranges.*

“The neuromorphometrics atlas seems to be an anatomical atlas? It is probably better to use a functional atlas given that you are looking at a measure of functional connectivity as there is large differentiation between anatomical and functional parcellations.”

Response: *We thank the reviewer for this comment. We have now re-run the colocalization analysis using the Schäfer 100 parcellation combined with another connectivity based parcellation (Melbourne/Tian atlas, 16 parcels) for subcortical regions (resulting in a similar total number of parcels as neuromorphometrics) and tested for similarity/robustness of the findings between ABIDE 1 and 2 datasets. Overall, the results are rather similar across the atlases (s. figure below) indicating the robustness of our findings. However, the effects with the Schäfer/Tian parcellation appear less consistent across both ABIDE datasets. Whilst for the neuromorphometrics parcellation, the obtained co-localization profiles across both ABIDE datasets correlate with $r=0.82$ the correlation drops to $r=0.59$ with the combined Schäfer/Tian parcellation indicating that this parcellation is not optimal for the co-localization analyses.*

Importantly in this regard, there are various functional atlases which much better delineate functional connectivity (including the Schäfer and Tian atlas deployed here) than the Neuromorphometrics atlas. However, they are primarily delineating the functional connectivity borders (providing for example in the case of the Schäfer atlas optimum parcellation based on long- and short-range connectivity profiles). Here we focus on LCOR, which is primarily a measure of regional homogeneity of signal for which we are not aware of any specific parcellation that optimizes specifically for this metric providing both cortical and subcortical parcellation. Moreover, to our knowledge there are no studies demonstrating that functional connectivity-based parcellation provides anyhow more optimum parcellation for different neurotransmitter properties which is the second part of the equation when computing the co-localization and therefore equally contributing to the observed correlations.

In summary, whilst neuromorphometrics might be not the optimum choice, we also do not believe that there is currently any compelling evidence supporting the notion that functional connectivity-based parcellation atlases would provide a better representation for other resting-state non-long range connectivity metrics or transfers better to underlying neurotransmission.

We have now added the following abbreviated version of the findings with the Schäfer/Tian parcellation to our manuscript:

“The results were largely replicated when using the functional Schaefer + Melbourne/Tian parcellation (see Supplementary Table S8).”

We now explain this in the respective methods section (p.26):

“We further tested for the robustness of these findings to changes in parcellation using a functional connectivity-based atlas, incorporating the Schaefer (100 parcels for cortical regions) and the Melbourne/Tian parcellations (16 parcels for subcortical regions).”

	ABIDE1 (Neuromorphometrics)		ABIDE1 (Schaefer + Melbourne/Tian)		ABIDE2 (Neuromorphometrics)		ABIDE2 (Schaefer + Melbourne/Tian)	
	Spearman r	p-value	Spearman r	p-value	Spearman r	p-value	Spearman r	p-value
5HT1a	.1982	.1349	.2958	.0370*	.0492	.7213	.0666	.6374
5HT1b	-.2480	.0809	-.0418	.7552	-.4308	< .001***	-.2725	.0519
5HT2a	.0555	.6993	.1120	.3946	-.0002	.9980	.0999	.4326
5HT4	.1945	.0240*	.1027	.2857	-.0334	.7423	-.0512	.5984
SERT	-.0393	.6553	-.2443	< .001***	-.0245	.7982	-.1670	.0689
D1	-.2634	.0050**	-.2756	.0020**	-.2778	.0020**	-.1587	.1029
D2	-.3216	< .001***	-.3708	< .001***	-.3239	< .001***	-.2157	.0160*
DAT	-.1867	.0330*	-.3132	.0020**	-.1963	.0280*	-.2361	.0090**
FDOPA	-.2000	.0340*	-.2075	.1259	-.0794	.3996	-.0256	.8581
NMDA	-.3318	.0090**	-.2989	.0170*	-.3109	.0090**	-.1430	.2817
mGluR5	-.3786	.0040**	-.2378	.0120*	-.3923	.0030**	-.1882	.0400*
GABAa	-.2478	.0050**	-.0613	.5315	-.1864	.0350*	.0716	.4446
CB1	-.1307	.3816	.0462	.7483	-.3348	.0110*	-.1775	.1958
MU	-.1056	.4905	-.0184	.9181	-.1484	.3400	-.1387	.3856
NAT	-.2018	.0240*	-.1797	.0569	-.1641	.0639	.0250	.7752
VACHT	-.2409	.0030**	-.2903	.0020**	-.2378	.0060**	-.2772	.0030**

Supplementary Table S8. Statistical data of co-localizations between LCOR alterations in autism compared to TD controls and neurotransmitter properties using the anatomical Neuromorphometrics atlas and the functional connectivity-based Schaefer + Melbourne/Tian atlas. ABIDE = autism brain imaging data exchange; TD = typically developed controls; 5HT = 5-hydroxytryptamine (serotonin receptor); SERT = serotonin transporter; D1/D2 = dopamine receptor; DAT = dopamine transporter; FDOPA = fluorodopa; NMDA = N-Methyl-D-Aspartat receptor; mGluR5 = metabotropic glutamate receptor; GABAa = γ -aminobutyric acid type A receptor; CB1 = cannabinoid receptor; MU = μ -opioid receptor; NAT = noradrenaline transporter; VACHT = vesicular acetylcholine transporter; * = $p < .05$; ** = $p < .01$; *** = $p < .001$.

Figure (for revision). Comparison of the co-localization fingerprints obtained across ABIDE1 and 2 with different atlases.

Reviewer: 2

“This is a very interesting paper using combined virtual functional histology merged with information from different drugs to explore whether functional differences in autism are associated with specific neurotransmitter systems which were subsequently tested. While this is an interesting paper, I do think the paper has some significant issues.”

Response: We sincerely thank the reviewer for their positive feedback on our manuscript. In response to the reviewer's comments, we have carefully revised the manuscript to address all points raised.

“First of all, the language is extremely pathologising. Recognising recent papers in both the US and UK (e.g. Bottema-beutel and colleagues) about preferred language, as well as recent studies showing the benefit of neuro-affirmative language in reducing stigma (compared to a deficit heavy language as used here), are really critical here. While I can accept that authors may prefer to use "DSM-5" language, they are othering neurodivergent individuals (incl. myself) in their language and should, at the least, address this in the intro or a footnote. This is quite simple, using Autism, instead of ASD. One approach could be "We are working on Autism Spectrum Disorder, henceforth referred to as "Autism"". Similarly, "participants" is much nicer language than the old school "subjects", and try to avoid "deficits", "dysfunction" and the like. Ultimately, this work is done FOR the community, so using community preferred language could be considered.”

Response: We thank the reviewer for this thoughtful comment and fully agree with their observation. We are committed to avoiding the pathologization or stigmatization of individuals with psychiatric diagnoses and are mindful of the significance of language, particularly in the context of autism. In response, we have carefully revised the manuscript to ensure the language is free from potentially pathologizing terms. Specifically, we have replaced “autism spectrum disorder” or “ASD” with “autism” and “autistic individuals.” Additionally, we have substituted terms like “impaired”, “deficits” and “dysfunction” opting instead for more neuro-affirmative language such as “differences,” “difficulties,” and “atypical”. Additionally, we have updated the manuscript title to:

“Local activity alterations in **individuals with autism** correlate with neurotransmitter properties and ketamine induced brain changes”

“Secondly, the big issue here is that the autistic brain responds different to drugs (as evidenced by many single drug probe studies, and recently shown in this preprint too: <https://www.medrxiv.org/content/10.1101/2024.11.08.24316969v1>) AND the "many ways lead to Rome" notion that different biological differences in autism lead to similar behavioural phenotypes. In other words, person A might have an affected GABA transporter, another an NMDA receptor, which both lead to a similar imbalance in "E/I" (a non specific umbrella term)

and behaviour. This heterogeneity isn't captured, explored, or explained. The samples are not generalisable and the pharmacological group is very small and only consists of healthy male adults which significantly affects generalisability and is a weakness of this study.”

Response: We thank the reviewer for this insightful comment and absolutely agree with the reviewer regarding all of the mentioned issues on the heterogeneity of drug responses in autism and the issue of “many ways lead to Rome” in autism research. Importantly in this regard, it was not our intention to make any claims that our findings anyhow explain the autism phenotype at an individual level or are useful as is to predict treatment response. More specifically, all of our findings only apply at group-level autism data providing evidence for disturbed excitation/inhibition ratio being a contributing factor to these alterations and these effects at group level are more similar to the effects induced by ketamine rather than midazolam in healthy individuals. We absolutely agree with the reviewer that these findings likely only represent a proportion of individuals with autism whilst many individuals in the autistic population have likely potential other effects being responsible for their phenotype. Nonetheless, they provide compelling in vivo evidence that these effects are relevant and we need to better understand in which individuals with autism this effect is most pronounced and potentially clinically relevant. Therefore, we also absolutely agree with the reviewer that understanding the heterogeneity in autism is a necessary step to develop personalized pharmacological interventions. However, we do not believe that the ABIDE datasets are an appropriate resource in this regard as the quality of the imaging data as well as the recruited clinical populations are extremely variable. Only group-level inference is therefore reliably possible in this cohort. We have now acknowledged this limitation of our study and also added the provided references:

“Noteworthy, distinct biological alterations, such as elevated glutamate neurotransmission or disrupted GABAergic signaling, can independently contribute to an increased E/I ratio in the brains of individuals with autism. These diverse mechanisms may ultimately converge to produce similar autistic phenotypes, complicating the interpretation of the underlying neurobiological pathways driving our observed results (Hollesetin et al., 2023; Litman et al., 2024)”^{7,8} (p.19); and

“Finally, it should be emphasized that the autistic brain may exhibit altered responses to pharmacological agents due to differences in neurotransmitter system responsivity (Whelan et al., 2024). This potential variability limits the generalizability of our pharmacological dataset, which was further constrained by the inclusion of only male participants and the impossibility to systematically account for the potential influence of prescribed medication on our findings”⁹ (p. 20).

“P4; It's not immediately relevant how rs-fMRI can support diagnosis? MRI's are expensive and there currently are no biomarkers at all for autism and most studies are mere case control. Autism is defined behaviourally. I don't really think the authors are right in this statement and they should consider removing it. I would argue rs-fMRI can aid in understanding the nature

of behavioural manifestations as part of the autistic phenotype. Beyond that the authors actually mostly look at case control differences, and they do not address heterogeneity well. This also refers to the discussion (p17) discussing different outcomes of clinical trials.”

Response: We apologize if we created the impression that we proposed rs-fMRI as being relevant for diagnosis of autism. We absolutely agree with the review, that based on the state-of-the-art rs-fMRI is still far away with respect to any possible applications for diagnosis of autism or actually any other psychiatric indications. We have now revised the respective statement accordingly. It now reads as follows (p.4):

“Resting-state functional magnetic resonance imaging (rs-fMRI) has the potential to offer objective insights into the neurophysiology of autism.”

“I would also disagree that the focus to date has been on glutamate. The focus has been on the E/I imbalance theory (which should be mentioned first) and in fact, predominantly on GABA.”

Response: We thank the reviewer for this suggestion. We have revised the respective paragraph as suggested putting the initial focus on the E/I imbalance starting the respective introduction paragraph with (p.5):

“The imbalance of excitation and inhibition (E/I) is a common mechanism discussed in relation to autism. In line with that, (...)”

“This sentence comes out of nowhere: "It also remains uncertain whether pharmacologically induced E/I modulation as elicited by glutamatergic medication ketamine and GABAergic medication midazolam in healthy subjects results in macroscale functional reorganization akin to that observed in ASD27–29." and generally I find the introduction poorly written and justified.”

Response: We thank the reviewer for highlighting this indeed misleading statement. We have now revised this statement to make a stronger connection to the previous paragraph. Together with the other changes that we implemented in response to the suggestions of this and the other reviewers, we hope that the introduction now reads more clearly and justified (p.5):

“Despite an increasing body of evidence identifying autism-related molecular and functional alterations, the relationship between both types of changes remains poorly understood. A viable way of addressing this question may be by modifying the E/I balance using glutamatergic or GABAergic drugs to modulate either excitation or inhibition respectively. One could then directly test whether such modulation induces functional changes that are similar to alterations observed in autism providing *in-vivo* evidence on the similarity of both effects.”

“Figure 1 and 2 could do with stats on the plot for interpretability.”

Response: As suggested by the reviewer, we have now added the stats to the respective plots.

“In the style of writing for Nature comms, I think the results could do with a bit more context on the samples for the two drugs and as above, motivation, details as this comes out of nowhere.”

Response: We appreciate the reviewer’s suggestion and agree that providing more context on the sample for the two drugs would enhance the clarity and motivation of our results. We have now added more details to the respective paragraph which now reads as follows (p22-23):

“To investigate the relationship between molecular brain signatures and aberrant local functional activity in autism through modulation of the E/I balance in health, we analyzed data from 30 healthy male participants without a history of recreational drug use. This cohort was originally collected to investigate the effect of drug manipulation on EEG and rs-fMRI metrics in healthy volunteers and the assessment procedures have been extensively described in Forsyth et al. (2020). Briefly, participants were scanned in a single-blinded, placebo-controlled, three-way crossover design, receiving ketamine, midazolam and placebo. The substances were administered to a sub-anesthetic level through intravenous access by an infusion pump. Female subjects were excluded to avoid variability in GABA levels associated with the menstrual cycle (Epperson et al., 2002). Three participants were excluded following image quality control, so that the final dataset consisted of 27 participants (M = 26.6 ± 5.9 years, 19-37 years).”^{10,11}

Furthermore, we have expanded the motivation for studying E/I alterations underlying autism by modifying brain states through glutamatergic and GABAergic medication (p.5):

“The imbalance of excitation and inhibition (E/I) is a common mechanism discussed in relation to autism. (...). A viable way of addressing this question may be by modifying the E/I balance using glutamatergic or GABAergic drugs to modulate either excitation or inhibition respectively. One could then directly test whether such modulation induces functional changes that are similar to alterations observed in autism providing *in-vivo* evidence on the similarity of both effects.”

“This statement does not seem supported by data at all: "This suggests that decreased LCOR in DMN nodes may relate to difficulties of individuals with ASD in self-referential processing and inferring mental states of others (“theory of mind”). Our findings further support recent efforts to identify neuro-subtypes in ASD, which have observed converging abnormalities in both the DMN and frontoparietal networks across subgroups 41,42." and the discussion is generally poorly written, reading more like a lit summary rather than contextualising the findings.”

Response: We apologize if this argument was unclear and acknowledge that it may have been a bit of overinterpretation of our results. The reviewer is correct in pointing out that this argument extends beyond what our data can support. Our intention was to emphasize the importance of the DMN in the pathophysiology of autism, a point that was further highlighted by our voxel-wise comparison of LCOR between individuals with autism and TD controls. This interpretation is also consistent with studies showing DMN alterations across different autism subtypes. In summarizing findings from task-based fMRI studies, we aimed to draw connections between reduced local functional activity within the DMN and impaired social cognition in autism. We have revised this section to make our argument clearer and more precise (p.15-16):

“Our results support the notion of consistent group-level local activity decreases as measured using LCOR in brain regions implicated in self-referential processing, social cognition, cognitive and emotional regulation. These findings align with recent rs-fMRI meta-analyses indicating local hypoactivity within the PCC⁶, precuneus and right temporal gyri and reinforce the role of functional abnormalities of the DMN in the pathophysiology of autism. Moreover, other studies utilizing ABIDE or other datasets have reported comparable alterations in regional homogeneity (ReHo), a metric closely related to LCOR (Ren et al., 2023; Zhao et al., 2022). The central role of the DMN in the pathophysiology of autism has been further emphasized by recent efforts to identify neurobiological subtypes of autism, which have observed converging abnormalities in both the DMN and frontoparietal networks across subtypes. Additionally, task-based fMRI studies have shown reduced activation in DMN regions during tasks requiring self-related versus other-related judgements. Taken together, these findings suggest that decreased functional activity within DMN nodes may be related to the challenges individuals with autism face in self-referential processing and theory of mind.”^{1,2}

“This is making the findings limited and not generalisable: "We excluded subjects with intellectual disability ($IQ \leq 70$) to reduce variability associated with low-functioning ASD" "Low functioning" is not a preferred term, this is absolutely a spectrum (see recent discussions surrounding profound autism). I would argue this is not about ASD but simply about inclusion and exclusion of intellectual disability. But it limits generalisability.”

Response: We thank the reviewer for this important observation and acknowledge that the term “low-functioning ASD” is outdated and no longer preferred. We have revised the manuscript to use more appropriate and inclusive language.

Regarding the exclusion of participants with an $IQ \leq 70$, we recognize that this criterion may limit the generalizability of our findings to individuals with autism who also have intellectual disability. However, our rationale for this exclusion was to reduce the potential confounding effects of intellectual disability on local functional activity measures. We have clarified in the revised manuscript that this exclusion criterion reflects a methodological decision to isolate the neural signatures associated with autism without the additional variability from intellectual disability, rather than implying any stratification within autism itself (p.22).

“We excluded subjects with intellectual disability ($IQ \leq 70$) because of its potential confounding effect on functional activity, (...).”

We have also added a statement to the discussion on this exclusion criterion limiting the generalizability of our findings (p.20):

“Furthermore, the generalizability of our findings is limited by the exclusion of autistic individuals with intellectual disability.”

“The drug datasets are extremely small and only consist of healthy males which again, limits generalisability, does not inform on autism and I would say is a significant downside of this study. It doesn't take into account gender effects at all and as such is an outdated paradigm.”

***Response:** We thank the reviewer for highlighting the limitations of our drug dataset. We acknowledge that the dataset is small and consists solely of healthy male participants, which indeed limits the generalizability of our findings. This design reflects the constraints of the original pharmacological study (Forsyth et al., 2020)¹⁰, which was intended as a controlled investigation of the acute effects of drug modulation on EEG and rs-fMRI metrics rather than as a direct study of autism. Importantly, the disadvantage of a smaller dataset is largely compensated by a within-subject cross-over design adopted in this dataset allowing to compare any effects of the drugs to the within-subject placebo response. We have now clarified this point in the manuscript to avoid any potential misinterpretation of the scope and relevance of this dataset to autism-specific research (p.22):*

“This cohort was originally collected to investigate the effect of drug manipulation on EEG and rs-fMRI metrics in healthy volunteers and the assessment procedures have been extensively described in Forsyth et al. (2020).”

Additionally, we have added a sentence explaining the reason for excluding female subjects from the original pharmacological study (p.22-23):

“Female subjects were excluded to avoid variability in GABA levels associated with the menstrual cycle (Epperson et al., 2002).”¹¹

We also recognize the importance of considering gender effects, particularly given the known sex-based differences in autism presentation and neural function. While our current dataset does not allow for this analysis, we acknowledge in the discussion that the exclusion of female participants represents a limitation of the study (p.20):

“Finally, it should be emphasized that the autistic brain may exhibit altered responses to pharmacological agents due to differences in neurotransmitter system responsivity (Whelan et al., 2024). This potential variability limits the generalizability of our pharmacological dataset, which was further constrained by the inclusion of only male participants and the impossibility to systematically account for the potential influence of prescribed medication on our findings.”⁹

Additionally, we have revised the conclusion section to explicitly highlight the need for future studies that incorporate more diverse samples, including participants with autism and balanced gender representation, to better explore how these factors may influence functional activity and neurotransmitter co-localization (p.20-21):

“Future neuro-subtyping studies with balanced gender representation aimed at disentangling the heterogeneity within autism will be crucial for identifying biologically and clinically distinct subgroups that allow for precision psychiatry approaches in autism.”

“Its not clear how the ADOS was used.”

Response: *We thank the reviewer for highlighting the need for further clarification on the use of the ADOS. As described in the methods section, we used the ADOS as a measure of autism symptom severity, focusing on its three core subdomains: communication difficulties, reciprocal social interaction (RSI), and stereotyped behaviors and restricted interests (SBRI). Additionally, to further investigate potential relationships between neurochemical alterations and specific symptom domains, we calculated Pearson correlations between the severity of symptoms in each subdomain and the strength of the co-localization between LCOR and respective neurotransmitter distributions in the brain. This analysis allowed us to explore how variations in neurotransmitter systems may be linked to distinct symptom profiles. We have now clarified this in the methods section (p.28):*

“We correlated the symptom severity of each subdomain with the individual Fisher’s z-transformed coefficients to explore how variations in neurotransmitter systems may be linked to distinct symptom profiles in autism.”

We appreciate the reviewer’s insightful feedback in helping us improve the transparency and rigor of our work. We hope these revisions provide the necessary clarification, and we appreciate the reviewer’s valuable feedback.

Reviewer: 3

“Summary: This article presents a sophisticated and novel approach to the E/I imbalance theory of autism. Researchers have capitalized on the large ABIDE I and II datasets, conducted a smaller scale medication trial of two agents that manipulate Glutamate and GABA systems, and compared spatial locations of hyper and hypo activation in autism to the pharmacological study results and to molecular imaging data. The findings are likely to have an impact on the field, spurring more such pharmacological trials for comparison to large databases. I have a few comments to improve the manuscript:”

Response: We thank the reviewer for this positive feedback on our manuscript. We have now thoroughly revised the manuscript complying with the reviewer’s suggestions.

“1. The phrasing of the sentence on page 5, last paragraph, “...show consistent local hyper- and hypoactivity in sensorimotor and DMN areas...” can be read in at least two ways with different meanings. Either it reads that the sensorimotor area is hyperactive while the DMN is hypoactive (likely intended meaning), or that each area is both hyperactive and hypoactive. The later meaning doesn’t really make sense, so I suggest re-writing this to emphasize which area(s) exhibit which findings.”

Response: We thank the reviewer for this helpful observation regarding the phrasing of this sentence. We acknowledge that the original wording may have been ambiguous. Our intended meaning was that sensorimotor areas exhibited local hyperactivity, while DMN regions showed local hypoactivity. To address this, we have revised the sentence for clarity to explicitly state (p.5-6):

“We hypothesize that (1) autistic individuals show consistent local hyperactivity in sensorimotor and hypoactivity in DMN areas that co-localize with the spatial distribution of specific neurotransmitter (...).”

We hope this revision eliminates any potential confusion, and we appreciate the reviewer’s feedback in helping us improve the precision of our manuscript.

“2. Typically, one discards the first 8 seconds or so of EPI fMRI data (sometimes referred to as “dummy” scans) so that the remaining scans represent a steady-state. I do not see mention of this. The authors should clarify if this was done, or provide a defense of not implementing it if not done.”

Response: We thank the reviewer for highlighting this important point. In response, we repeated the preprocessing of the functional imaging data by discarding the first four scans (~8 seconds) of each fMRI run to ensure that the remaining data reflected a steady-state signal. Additionally, as noted by Reviewer #1, we removed both smoothing and gray matter signal regression from the preprocessing pipeline due to their potential influence on LCOR. The

updated analyses revealed robust, albeit less extensive, significant LCOR reductions in individuals with autism compared to TD controls (see Reviewer #1, Comment 4):

“The voxel-wise findings were robust but spatially less extended following changes in preprocessing pipeline including removing the first four initial scans, smoothing and gray matter signal regression (Supplementary Figures S1-2).” (p.7)

Furthermore, the co-localization profiles for ABIDE1 and ABIDE2 remained stable, and the co-localization with mGluR5 and GABA_A receptors was even stronger under these conditions (p.8):

“The co-localization profiles were robust to changes in pre-processing pipeline including removing the first four initial scans and omitting smoothing and gray matter signal regression (see Supplementary Tables S6-7).”

Additionally, we explained this procedure in the methods section (p.24):

“Given that both spatial smoothing and gray matter signal regression can have a significant impact on functional activity measures and subsequent analyses (Herrera et al., 2016; Triana et al., 2020; Vos de Wael et al., 2017), we evaluated the robustness of our primary results by excluding these preprocessing steps from the pipeline. Additionally, we removed the first four initial scans for each subject to further ensure the stability of our findings.”³⁻⁵

We appreciate the reviewer’s thorough review and the opportunity to clarify these aspects of our preprocessing pipeline.

“3. Page 25, top sentence: provide citation for Benjamini-Hochberg FDR procedure.”

***Response:** We thank the reviewer for pointing out the need for a citation for the Benjamini-Hochberg FDR procedure. We have now included the appropriate reference to the original publication (Benjamini & Hochberg, 1995) in the revised manuscript. We appreciate the reviewer’s attention to detail in improving the rigor of our manuscript.*

“4. If drug effects were expected to produce changes in both directions (ASD>TD or TD>ASD) on calculated delta scores, why were one-sided tests conducted? I’m not sure this is justifiable.”

***Response:** We thank the reviewer for raising this important point regarding the use of one-sided tests. Our hypothesis was that both ketamine and midazolam would induce autism-like alterations in local functional activity, specifically mirroring the directionality of LCOR changes observed in autism relative to TD controls. Although ketamine and midazolam primarily affect different brain regions — consistent with their distinct pharmacological actions on NMDA and GABA_A receptor systems — we hypothesized that both drugs would induce autism-related LCOR alterations in the regions identified as significantly different*

between autism and TD groups. Given the clear directional nature of our hypothesis, we used one-sided tests to increase the sensitivity of our analysis to detect these specific shifts.

“5. In the supplementary figures S2 and S3, the threshold used for visualization of the voxel-wise results should be stated in the figure notes.”

Response: *We thank the reviewer for their attention to this detail. We have now added the threshold for voxel-wise comparisons in the notes of these Supplementary Figures (which are now labeled as S4 and S5) to ensure transparency.*

“6. In the discussion of mGluR5, it would be worth referencing the interest in this as a treatment target in Fragile X Syndrome, since that disorder is one of a few single gene disorders associated with autism.”

Response: *We thank the reviewer for this important suggestion. Fragile X Syndrome is indeed one of the well-characterized single-gene disorders associated with autism, and mGluR5 has been extensively studied as a potential treatment target in this context. We have now incorporated a discussion of this point in the revised manuscript, highlighting the relevance of mGluR5 in Fragile X Syndrome and its potential implications for autism research more broadly (p.17):*

“Fragile X syndrome, the most common known single-gene cause of autism, is also linked to dysregulation of mGluR5 signaling, and pharmacological mGluR5 antagonists have demonstrated promising effects in preclinical and early clinical studies (Brašić et al., 2021; Pop et al., 2013).”^{12,13}

We appreciate the reviewer’s valuable input in strengthening the contextualization of our findings.

Reviewer: 4

“This study looks at how brain activity differs in people with autism spectrum disorder (ASD) and what might be causing those differences. Researchers analyzed brain activity at rest in people with ASD and compared it to people without ASD using data from two large groups. They found that people with ASD had consistently lower brain activity in certain areas, especially in parts of the brain involved in thinking about oneself and others (the default mode network).

These changes in brain activity were linked to differences in several brain chemical systems, including those involving dopamine, glutamate, GABA, and acetylcholine—essential chemicals that help brain cells communicate. The researchers also tested how certain drugs affect brain activity. They found that ketamine, which affects glutamate, caused brain changes similar to those seen in ASD, while midazolam, which affects GABA, did not.

This suggests that disruptions in the balance of brain chemicals might play a role in the brain differences seen in ASD, helping us better understand its underlying causes and offering candidate biomarkers to inform treatment development.

Strengths of the work include:

1. Using the ABIDE datasets provides for easy replication of the analysis and extension of the work by other researchers.
2. The test-retest approach in ABIDE 1 vs. ABIDE 2 is very nice.
3. The impact of the findings is high. The work will likely lead to new avenues for treatment development in ASD.
4. While focused on autism, the work has broad implications for our understanding of brain function and neurological conditions broadly.
5. The use of a placebo control, as well as a comparison of two different drugs targeting two different receptor systems, in PET imaging is compelling.”

***Response:** We thank the reviewer for this very positive feedback on our manuscript. We have now thoroughly revised the manuscript complying with the reviewer’s suggestions.*

“Minor weaknesses are:

1. The authors confused me a bit in setting the stage in the introduction. I suspect this was because of the need to be concise for this journal’s format. Specifically, I know why it is fascinating that the changes in ASD may mirror those induced by ketamine and/or midazolam, but presently, this isn’t well articulated in the introduction. This clarification will help a broader audience and ASD-focused audience appreciate the importance of the work. To keep the manuscript concise, you could take a bit from the excellent discussion of this point from the

discussion section and move it here or duplicate in both places to ensure the reader follows the motivation clearly before getting into the results.”

Response: *We thank the reviewer for this valuable feedback. We appreciate the importance of clearly articulating the rationale behind investigating ketamine- and midazolam-induced changes in relation to autism-related alterations. To improve clarity for both a broader audience and autism-focused readers, we have revised the introduction to more explicitly outline why these pharmacological models are relevant to autism research. Specifically, we have more directly introduced the relevance of E/I imbalance in autism (p.5):*

“The imbalance of excitation and inhibition (E/I) is a common mechanism discussed in relation to autism. In line with that, alterations in glutamatergic neurotransmission have been proposed as pivotal in the neurophysiology of autism.”

We further clarify how ketamine and midazolam modulate the E/I ratio and why this is relevant to our approach (p.5):

“Despite an increasing body of evidence identifying autism-related molecular and functional alterations, the relationship between both types of changes remains poorly understood. A viable way of addressing this question may be by modifying the E/I balance using glutamatergic or GABAergic drugs to modulate either excitation or inhibition respectively. One could then directly test whether such modulation induces functional changes that are similar to alterations observed in autism providing *in-vivo* evidence on the similarity of both effects.”

These revisions ensure that the motivation behind our study is clearly established before presenting the results. We appreciate the reviewer’s suggestion, which has helped enhance the readability and impact of our manuscript.

“2. Please see the questions below. Depending on your responses, some of these questions could be points that you need to consider as potential limitations to be discussed in contextualizing your findings.

Questions for the authors:

1. To what degree are subjects independent across ABIDE 1 and 2? Are some subjects, likely at different points in development, duplicated in each collection?”

Response: *We thank the reviewer for raising this important point. ABIDE1 and ABIDE2 are independent data collections, and while some imaging sites contributed to both datasets, they did not include the same individuals at different time points. Thus, there is no subject duplication across ABIDE 1 and ABIDE 2 in our analyses.*

To ensure data independence, we carefully reviewed the participant identifiers and metadata provided by ABIDE and confirmed that all subjects included in our study were unique.

We appreciate the reviewer's attention to this detail.

“2. What information is available regarding medications taken by subjects in the ABIDE collections? I ask because GABA-, Dopamine-, and Serotonin-targeting medications are commonly prescribed for associated conditions in autistic patients.”

Response: *We appreciate the reviewer's important question. Medication information is unfortunately only partially available in the ABIDE dataset. The level of detail strongly varies across sites, and comprehensive data on specific GABA-, dopamine-, and serotonin-targeting medications is not consistently reported. Given these limitations, we were unable to systematically account for the potential influence of medication on our findings.*

We now acknowledge this as a limitation of our study and have explicitly stated this in the discussion (p.20):

“This potential variability limits the generalizability of our pharmacological dataset, which was further constrained by the inclusion of only male participants and the impossibility to systematically account for the potential influence of prescribed medication on our findings.”

Future studies with more detailed pharmacological data could help disentangle the potential effects of commonly prescribed medications on local functional activity and neurotransmitter co-localization in autism.

“3. Related to the question above, is information available about the presence of seizures for the ABIDE collections?”

Response: *We thank the reviewer for this question. While epilepsy and seizures are known to be more prevalent in autism, information on the presence of seizures is not available in the ABIDE datasets. Due to this limitation, we were unable to account for the potential influence of seizure history on our findings.*

We acknowledge this as a limitation and have now noted it in the discussion (p.20):

“In that regard it is also important to note that the included multi-site datasets are extremely heterogeneous with respect to availability of demographic, medication status and clinical information (e.g., seizure history) as well as in terms of image quality.”

“4. In line 94, should “neurotransmitter” be plural here?”

Response: *We have now corrected the neurotransmitter to plural.*

“5. Are the Social Responsiveness Scale scores for participants available in ABIDE? If so, this might be a more powerful tool to examine behavioral correlates of the observed activation patterns.”

Response: *We thank the reviewer for this suggestion. Social Responsiveness Scale (SRS) scores were available for a subset of participants with autism in the ABIDE datasets (ABIDE 1: $n = 167$; ABIDE2: $n = 313$) and the mean total score was comparable between both cohorts (ABIDE1: $M = 85.34$, $SD = 36.44$; ABIDE2: $M = 90.96$, $SD = 26.67$). However, our analysis did not reveal any significant correlation between individual Fisher z-scores (neurochemical co-localization) and the SRS total score or any subscore across both cohorts (see Table below)*

We note that the number of participants with available SRS subscores was relatively small in ABIDE1 (~30 participants), which may have limited the comparability between both cohorts and the statistical power to detect meaningful associations with distinct behavioral autism phenotypes.

We appreciate the reviewer's input and acknowledge that future studies with larger, well-characterized cohorts may be better suited to investigate potential links between neurochemical co-localizations and behavioral phenotypes in autism.

	SRS	ABIDE1		ABIDE2	
		r	p	r	p
D1	Total	.007	.925	.025	.656
	Awareness	.341	.052	-.033	.582
	Cognition	.136	.436	-.064	.280
	Communication	.195	.276	.034	.554
	Motivation	.155	.389	.061	.280
	Mannerisms	.473	.015*	-.029	.612
D2	Total	-.021	.784	.069	.221
	Awareness	.115	.526	.094	.112
	Cognition	.075	.669	-.049	.412
	Communication	.069	.704	.089	.115
	Motivation	.008	.965	.086	.129
	Mannerisms	.143	.587	.035	.538
DAT	Total	-.042	.590	.059	.296
	Awareness	.159	.376	.040	.499
	Cognition	.147	.401	-.026	.665
	Communication	.104	.564	.071	.211
	Motivation	.111	.539	.063	.264
	Mannerisms	.263	.195	-.011	.852
NMDA	Total	.018	.816	.057	.319
	Awareness	.181	.314	.062	.297
	Cognition	.115	.509	-.048	.418
	Communication	.099	.582	.062	.277
	Motivation	.045	.805	.098	.084
	Mannerisms	.227	.266	.014	.807
mGluR5	Total	.036	.643	-.041	.468
	Awareness	.039	.828	.016	.785
	Cognition	-.237	.171	-.064	.280
	Communication	-.143	.429	-.021	.716
	Motivation	-.150	.404	.002	.975
	Mannerisms	.119	.563	-.042	.459
GABAa	Total	.022	.780	-.037	.519
	Awareness	.109	.545	.023	.694
	Cognition	-.109	.532	-.053	.376
	Communication	-.028	.878	-.013	.816
	Motivation	-.146	.416	-.004	.945
	Mannerisms	.286	.156	-.051	.370
VACHT	Total	-.035	.651	.051	.368
	Awareness	.128	.477	.033	.578
	Cognition	-.055	.754	-.014	.814
	Communication	-.054	.764	.053	.354
	Motivation	.000	.998	.092	.104
	Mannerisms	-.014	.946	.032	.572

Figure (for revision). Pearson correlations between the strength of LCOR-neurotransmitter co-localizations in autism and the Social Responsiveness Scale (SRS) total score and subscales. * = $p < .05$.

“In sum, this is an exciting, impactful study. The paper is well-written and compelling. The conclusions closely follow from the results. The work will be of interest to a broad audience.”

***Response:** We sincerely thank the reviewer for their positive assessment of our study. We greatly appreciate their thoughtful evaluation and are pleased to hear that they found the work exciting, impactful, and relevant to a broad audience. Their constructive comments have been invaluable in further refining the manuscript.*

References

1. Ren, P. *et al.* Stratifying ASD and characterizing the functional connectivity of subtypes in resting-state fMRI. *Behav. Brain Res.* **449**, 114458 (2023).
2. Zhao, X. *et al.* Regional homogeneity of adolescents with high-functioning autism spectrum disorder and its association with symptom severity. *Brain Behav.* **12**, 1–12 (2022).
3. Herrera, L. C. T., Castellano, G. & Coan, A. C. Impact of gray matter signal regression in resting state and language task functional networks. 1–28 (2016) doi:10.1101/094078.
4. Triana, A. M., Glerean, E., Saramäki, J. & Korhonen, O. Effects of spatial smoothing on group-level differences in functional brain networks. *Netw. Neurosci.* **4**, 556–574 (2020).
5. Vos De Wael, R., Hyder, F. & Thompson, G. J. Effects of Tissue-Specific Functional Magnetic Resonance Imaging Signal Regression on Resting-State Functional Connectivity. *Brain Connect.* **7**, 482–490 (2017).
6. Dukart, J. *et al.* JuSpace: A tool for spatial correlation analyses of magnetic resonance imaging data with nuclear imaging derived neurotransmitter maps. *Hum. Brain Mapp.* **42**, 555–566 (2021).
7. Hollestein, V. *et al.* Excitatory/inhibitory imbalance in autism: the role of glutamate and GABA gene-sets in symptoms and cortical brain structure. *Transl. Psychiatry* **13**, 1–9 (2023).
8. Litman, A., Sauerwald, N., Snyder, L. G., Foss-feig, J. & Christopher, Y. Decomposition of phenotypic heterogeneity in autism reveals distinct and coherent genetic programs. (2024).
9. Whelan, T. P. *et al.* Autistic Neural Shiftability: The Distinct Pharmacological Landscape of the Autistic Brain. *medRxiv* 1–33 (2024) doi:10.1101/2024.11.08.24316969.
10. Forsyth, A. *et al.* Modulation of simultaneously collected hemodynamic and electrophysiological functional connectivity by ketamine and midazolam. *Hum. Brain Mapp.* **41**, 1472–1494 (2020).
11. Neill Epperson, C. *et al.* Cortical γ -aminobutyric acid levels across the menstrual cycle in healthy women and those with premenstrual dysphoric disorder: A proton magnetic resonance spectroscopy study. *Arch. Gen. Psychiatry* **59**, 851–858 (2002).
12. Brašić, J. R. *et al.* Cerebral expression of metabotropic glutamate receptor subtype 5 in idiopathic autism spectrum disorder and fragile x syndrome: A pilot study. *Int. J. Mol. Sci.* **22**, 1–16 (2021).
13. Pop, A. S., Gomez-Mancilla, B., Neri, G., Willemsen, R. & Gasparini, F. Fragile X syndrome: A preclinical review on metabotropic glutamate receptor 5 (mGluR5) antagonists and drug development. *Psychopharmacology (Berl)*. **231**, 1217–1226 (2014).

We thank the reviewers for providing their positive reviews of our manuscript. We have thoroughly revised and complied with the reviewers' suggestions, and we believe that the paper has substantially benefited from this review. Authors' point-by-point responses to the reviewers' comments are provided below and corresponding changes in the revised manuscript are highlighted in yellow color.

Reviewer #1:

“Reviewer comment 1: I appreciate the author's responses to my previous comments which they have addressed. However, this comment does not really address my concern which is that the interpretation remains unclear of how this results speaks to the relationship between autism and drug related effects. As mentioned in my original comment, two things can be correlated even if they are not related.

If I correlated subscales of the ADOS with subscales of some measure of depression and found they were related, this relationship would be questionable if neither the total ADOS nor the total depression scale passed the threshold for a clinical diagnosis. I am having a hard seeing how one should interpret the relationship between the two as there is no official diagnosis of either autism nor depression even though they are significantly correlated. It is the same in this situation.”

Response 1: *We thank the reviewer for this thoughtful comment on the provided correlational approach linking the autism phenotypes to drug related effects. First, we fully agree with the reviewer that any kind of statistical testing, including our chosen correlational approach, cannot demonstrate causality or exclude the possibility of spurious findings. Indeed, as the reviewer rightly points out, significant correlations may result from different possible mechanisms, including bidirectional causality or underlying confounding. Thus, we cannot make definitive claims regarding the nature of the observed associations. This was a major reason for us to aim to replicate any of our findings to the extent possible with existing large data resources. We demonstrate such a robust replication for the associations of autism alterations with specific neurotransmitter maps with all major association effects resulting in meta-analytic p-values of $p < .001$ (s. Supplementary Table S9). Whilst not demonstrating any causality the robust associations strongly point to the existence of biological mechanisms contributing to autism alterations to manifest in a network that significantly aligns among others with glutamatergic and GABAergic neurotransmitter systems. Importantly, all individuals with autism in the large ABIDE 1 and 2 cohorts were diagnosed with autism. We would therefore not expect these effects to be anyhow explainable by such subthreshold symptom manifestation as provided in the example of the reviewer. These replicable findings also provided for us a biological rationale to test if the spatial topologies of these effects are also similar to those induced by compounds that are known to directly modulate the respective neurotransmitter systems. Both ketamine and midazolam are strongly pharmacologically active and the doses for the study in healthy volunteers were chosen based on previous experiments*

demonstrating robust behavioral and neuroimaging effects at the respective drugs. Also here, we would therefore not expect that the observed effects are anyhow spurious due to insufficient dosing or similar.

Importantly, by linking autism and drug effects we do not anyhow imply any causality that for example NMDA imbalance is causing autism. We merely demonstrate that there is a likely common biological mechanism leading to topologically similar alterations induced by ketamine (and partially midazolam) to those observed in autism. And here again, to ensure the robustness of our findings we demonstrate this similarity in three different ways: 1) purely on regional level (following classical analyses for this type of data) showing that ketamine significantly reduces LCOR in regions that are also decreased in autism, 2) that ketamine effects co-localize with NMDA, GABA_A and other neurotransmitter distributions as also observed in autism and 3) that the spatial co-localization profiles of ketamine across all neurotransmitters are similar to those obtained in ABIDE 1 and 2 (providing additional replication). Whilst we agree with the reviewer that each analysis per se is only observational, the combined evidence across the different levels provides a compelling argument towards a common biological mechanism linking both types of observations.

Noteworthy, the drug effects, even if measured in a relatively small cohort (as pointed in the next comment of the reviewer), are established in one of the best standard study designs, namely in a randomized single-blinded placebo-controlled cross-over design comparing each subjects' ketamine and midazolam response to their own placebo response. This type of design is considered a gold standard for drug studies and offers substantial statistical advantages over cross-sectional designs. For example, Bellemare et al. (2014)¹ have shown that depending on the signal and noise assumptions, you need between 5 and 10 times less subjects to demonstrate the same significant effects. This basically means that our cross-sectional design in a conservative estimate would correspond to the statistical power of a cross-sectional comparison of about 150-200 subjects which would be considered a fairly large cohort. Moreover, as for these cohort we have also collected cerebral flow (CBF) and EEG data, we have previously shown that at the chosen dose both drugs also induce a robust CBF and EEG response further demonstrating the robustness of the observed pharmacodynamic effects^{2,3}.

Returning to the example raised by the reviewer, as discussed above, we would argue that neither autism effects nor the drug effects were derived from insufficiently sampled or subclinical/insufficiently dosed data. We would therefore consider both autism and drug effects as robustly established. In this regard and as outlined more in detail below, our study provides novel evidence in support of one of the major prevailing hypotheses of an E/I imbalance contributing to the occurrence of autism as our findings match fairly well what would be predicted from the E/I imbalance theoretical framework discussed quite extensively in the literature as a possible mechanism contributing to autism phenotypes.

With regard to the interpretation issue raised by the reviewer, we would like to emphasize again that we fully agree that our findings do not imply any causal interpretation such as “autism being caused by glutamatergic disturbance” or similar. Our findings are inherently limited, among other factors, by the cross-sectional and group-averaging study design — an aspect we now emphasize more clearly in the revised limitations section of the discussion (p.20):

“In that regard it is also important to note that the included multi-site datasets are extremely heterogeneous with respect to the availability of demographic, medication status and clinical information (e.g., seizure history) as well as in terms of image quality. This variability may obscure more nuanced insights into the neurobiological mechanisms underlying autism. Moreover, the cross-sectional and group-averaging nature of our study design precludes any causal interpretation of our findings. Future research should aim to conduct large-scale, standardized, and longitudinal investigations to enable the identification of distinct autism subtypes and their associated neurobiological profiles.”

“We apologize again for the unclear description of the applied methodology. The reviewer is absolutely correct that regions which do not display any autism (or drug related) effects would show random fluctuations in their response. However, first of all, those random fluctuations are in the first place random and therefore not expected to induce any significant correlations exceeding those observed in randomly generated maps with the same spatial smoothness. If only such random effects prevail the statistics would simply become not significant.

Reviewer comment 2: Correlation does not equal causation and just because two things are associated, it doesn't mean they are related as claimed, nor does it mean that the interpretation of their relationship is clear.”

Response 2: Again, we absolutely agree with the reviewer that correlation does not equal causation. We have also gone thoroughly through the manuscript and revised any possible statements that could be interpreted that we imply any causality of our findings (e.g. words like “produce” or “pathomechanism”; please see line 128, 257-258, 277, 298-299, 319-320). For example, statements in the abstract that could be interpreted as causal were modified (s.3).

“Autism is a neurodevelopmental condition associated with altered resting-state brain function. An increased excitation-inhibition (E/I) ratio is discussed as a potential pathomechanism but in-vivo evidence of disturbed neurotransmission related to these functional alterations remains scarce. We compared rs-fMRI local activity (LCOR) between autism (N=405, N=395) and neurotypical controls (N=473, N=474) in two independent cohorts (ABIDE1 and ABIDE2). We tested how these LCOR alterations co-localize with specific neurotransmitter systems derived from nuclear imaging and compared them with E/I changes induced by GABAergic (midazolam) and glutamatergic medication (ketamine). Across both cohorts, autistic individuals consistently exhibited reduced LCOR, particularly in higher-order default mode network nodes. The whole-brain LCOR alterations negatively co-localized with dopaminergic (D1, D2, DAT), glutamatergic (NMDA, mGluR5), GABAergic (GABA_A) and cholinergic neurotransmission (VACHT). The NMDA-antagonist ketamine, but not GABA_A-potentiator midazolam, induced LCOR changes which co-localize with D1, NMDA and GABA_A receptors, thereby resembling alterations observed in autism. We find consistent local activity alterations in autism spatially associated with several major neurotransmitter systems. Ketamine induced neurochemical changes similar to those seen in autism, suggesting that pharmacological modulation of the E/I balance in healthy individuals is associated with brain changes similar to those observed in autism. These findings provide novel insights which may contribute to our understanding of the neurophysiological basis of autism.”

The E/I imbalance theory/framework is one of the prevailing theories of autism. Assuming that this theory holds, we can make clear predictions to our data that would provide evidence in favor or against it. If autism effects are related to glutamatergic or GABAergic systems, we would expect the effects in autism to 1) co-localize with the known distribution of the respective

receptors across the brain, 2) compounds that modulate either the E or the I of E/I imbalance to induce similar spatial alterations as those observed in autism. We test both of these predictions in our data with particular attention paid to robustness and whenever possible replication of the respective findings. In that regard, our findings only provide evidence that the predictions that would be made from E/I imbalance framework are what we actually observe in our data. As for any scientific non-causal design, they do not demonstrate causality but merely provide cumulative statistical evidence.

"Second, the reason of using t-maps (as a measure of effect size) in our study and other similar spatial co-localization studies is exactly to minimize the contribution of this type of random fluctuations as those (considering the large sample size) will only generate weak effect sizes tending close to zero whilst mostly only true positive effects will induce larger positive and negative t-values and therefore contribute to the actual spatial correlations that deviate from a random distribution of such correlations.

Reviewer comment 3: The sample size is small for the drug effects causing concern of spuriously large associations."

Response 3: As the issue of sample size has been addressed in detail above, we would like to briefly reiterate and clarify our rationale in the interest of transparency and readability. While the pharmacological effects in our study were indeed assessed in a relatively small cohort, this was done using a well-established and rigorous design for pharmacodynamic effects, namely in a randomized single-blinded, placebo-controlled cross-over paradigm. Each participant's response to ketamine and midazolam was directly compared to their own placebo response, thus minimizing between-subject variability.

This within-subject cross-over design is widely regarded as a gold standard in pharmacodynamic research, offering notable statistical advantages over conventional cross-sectional designs. For example, Bellamare et al. 2014. (s. reference below) have demonstrated that, depending on the signal and noise assumptions, such designs can require 5 to 10 times fewer subjects to achieve comparable statistical power in a within as compared to a between subject design. This basically means that our cross-over design in a conservative estimate would correspond to the statistical power of a cross-sectional comparison of about 150-200 subjects which would be considered a fairly large cohort and pointing to a rather high robustness of the demonstrated pharmacodynamic alterations.

Moreover, in the same cohort we have also collected cerebral blood flow (CBF) and EEG measures following drug administration (Dukart et al. 2018, Forsyth et al. 2018). Here, we have shown that at the chosen dose both drugs also induce robust CBF and EEG changes in this dataset further supporting the reliability of the pharmacodynamic effects observed in the present study.

"Whilst it is conceptually possible to restrict the spatial co-localization approach to only regions which for example have significant effects in autism, this type of restriction creates other types of issues which completely outweigh its advantages. For example, correlation of only a preselection of autism-related regions with neurotransmitter maps may artificially narrow down the actual range of the neurotransmitter information, i.e. regions with only very low or very high expression of a specific receptor get selected. In the case of the low expression, the correlation in those regions will then appear meaningless as all selected regions basically do contain any meaningful receptor signal and the variance therefore mostly reflects noise. In the case of high expression, a true correlation may be lost as we only look at regions with similarly high expression and completely lose the contrast against regions which have low or moderate expression of the respective signal. As we have no valid ways to prevent or detect such thresholding induced biases without looking at the actual unthresholded maps, the direct use of unthresholded maps combined with proper non-parametric statistics appears more appropriate and rigorous.

Reviewer comment 4: While this approach offers more possibilities to find significant effects, that is not really a justification for forcing an interpretation on the results that is not supported. A region should be selected because it is related to autism and related to drug administration, not because selecting more regions offers more chances to find a result.

As in the above example I gave, that approach would necessitate that I examine only those with a diagnosis of autism and a diagnosis of depression, while this approach that is offered would examine all subjects results regardless of diagnosis. The former comparison is easily justified as looking at the relationship between autism and depression while for the latter it is harder to state exactly what the relationship is."

Response 4: We appreciate the opportunity to clarify this point and apologize again if the methodological description was not sufficiently clear. It appears there may be a conceptual misunderstanding regarding our chosen analytical approach.

The analysis suggested by the reviewer was indeed implemented as the initial step in linking the ketamine and autism data. Namely, we first look only at one region/cluster that was significantly altered in autism and tested (only one test performed per cluster and drug) if the signal there and only there is changed by either ketamine or midazolam. These results are reported in Figure 2a. This analysis directly addresses the comment of the reviewer as it demonstrates that autism-related regions are modified by both drugs. As outlined in our response during the previous revision of the manuscript, this approach was chosen over a conjunction type of analysis suggested by the reviewer because we aimed to test whether regions altered in autism are modulated by the both drugs and not whether there is a common subset of regions is altered/modulated by both. However, whilst this analysis demonstrates sensitivity it does not demonstrate specificity of the effects of both compounds to autism associated regions. For example, it could be that other regions that are not altered in autism would show very similar or stronger drug signals. For this, we performed spatial correlation analyses to demonstrate that the spatial topology of the drug effects is similar to the topology of the autism alterations across the whole brain. This correlation type of analysis explicitly tests across the whole brain if the effect of the drugs is stronger in regions altered by autism.

Importantly, and here we believe there is the misunderstanding, the significance of this analysis is not anyhow statistically dependent on selecting more or less regions as the significance is determined using permutation testing with the very same number of regions. It does also not increase the chance of finding significant results as we never test any single region or their combinations separately but always only perform one test per neurotransmitter map. Basically, if we select 100 regions, the observed true correlation is tested against permuted maps each with 100 regions and similar topological properties (i.e. preserving spatial auto-correlation).

In contrast, selecting only a subset of regions for this analysis (irrespective of the selection criteria) would be methodologically very ill suited for our question as we would not be able to demonstrate specificity in such a case. If we select only regions that are affected by autism and these regions align with NMDA distribution (i.e. only regions with high expression are altered in autism) then correlation within those regions becomes meaningless as they all carry a very similar signal of the neurotransmitter. Staying in the above example given by reviewer, such narrowing (preselection of regions) would equate to threshold two clinical scales to a subclinical level and then try to detect a correlation in this very narrowed and not clinically meaningful range. To detect a correlation that is meaningful we basically need to look at the full range of values and not to a randomly selected narrowed subset.

Reviewer comment 5: “I am unsure how one can state what the relationship is between the two variables used in this study in this particular way, but it doesn't seem like one can claim it is a relationship strictly between autism and drug administration.”

Response 5: We fully agree with the reviewer's important point. As now emphasized in our discussion, we do not draw any causal inferences from the observed correlations. Our analyses are based on cross-sectional data, and we correlate spatial topographies across independent datasets. Our aim was to assess whether the spatial pattern of functional alterations observed in autism aligns with the spatial distribution of neurotransmitter systems implicated in the E/I imbalance framework. In that sense, our findings provide converging and robust support for this model at the topographical level, but do not permit any conclusions about causality.

Reviewer #2:

Reviewer comment 1: “To the authors, I want to commend the authors on a thorough well thought out response to the reviewer comments. All the comments I made previously have been addressed appropriately and I appreciate the authors taking the time to significantly revise the manuscript. While I have some minor issues that cannot be addressed (e.g. dealing with mild ID, only males, and conceptualisation of the ADOS as an outcome) I don't believe any further discussion/revisions will benefit the manuscript. I am happy with this.“

Response 1: We thank the reviewer for their constructive feedback throughout the review process. We appreciate your helpful comments, which have contributed to strengthening the manuscript. We acknowledge the limitations you have noted and agree that these are important considerations for future work.

Reviewer #4:

Reviewer comment 1: “I am pleased with the revisions and answers to my questions. The paper is now in great shape.”

Response 1: We sincerely thank the reviewer for their feedback and are pleased that the revisions and responses have addressed your concerns. We appreciate your positive assessment and are grateful for your role in helping to improve the quality of the manuscript.

References

1. Bellemare, C., Bissonnette, L. & Krrger, S. Statistical Power of within and Between-Subjects Designs in Economic Experiments. *SSRN Electron. J.* (2018) doi:10.2139/ssrn.3149007.
2. Dukart, J. *et al.* Cerebral blood flow predicts differential neurotransmitter activity. *Sci. Rep.* **8**, 1–11 (2018).
3. Forsyth, A. *et al.* Comparison of local spectral modulation, and temporal correlation, of simultaneously recorded EEG/fMRI signals during ketamine and midazolam sedation. *Psychopharmacology (Berl)*. **235**, 3479–3493 (2018).